# SimXRD-4M: Big Simulated X-ray Diffraction Data and Crystal Symmetry Classification Benchmark

**Bin Cao**[1*] **Yang Liu**[2,4*] **Zinan Zheng**[2*] **Ruifeng Tan**[1,3], **Jia Li**[2,4†] **Tong-yi Zhang**[1,3†]

[1]Guangzhou Municipal Key Laboratory of Materials Informatics, Advanced Materials Thrust,
The Hong Kong University of Science and Technology (Guangzhou)
[2]Data Science and Analytics Thrust, The Hong Kong University of Science and Technology (Guangzhou)
[3]Sustainable Energy and Environment Thrust, The Hong Kong University of Science and Technology (Guangzhou)
[4]The Hong Kong University of Science and Technology

## Abstract

Powder X-ray diffraction (XRD) patterns are highly effective for crystal identification and play a pivotal role in materials discovery. Although machine learning (ML) has advanced the analysis of powder XRD patterns, progress has been constrained by the limited availability of training data and established benchmarks. To address this, we introduce SimXRD-4M, the largest open-source simulated XRD pattern dataset to date, aimed at accelerating the development of crystallographic informatics. We developed a novel XRD simulation method that incorporates comprehensive physical interactions, resulting in a high-fidelity database. SimXRD comprises 4,065,346 simulated powder XRD patterns, representing 119,569 unique crystal structures under 33 simulated conditions that reflect real-world variations. We benchmark 21 sequence models in both in-library and out-of-library scenarios and analyze the impact of class imbalance in long-tailed crystal label distributions. Remarkably, we find that: (1) current neural networks struggle with classifying low-frequency crystals, particularly in out-of-library situations; (2) models trained on SimXRD can generalize to real experimental data.

## 1 Introduction

Symmetry identification is a fundamental and crucial step for materials characterization and design. Specifically, Powder X-ray diffraction (XRD) analysis is exceptionally potent in probing microstructure, due to its sensitivity to atomic arrangement and the element specificity of atom scattering power. The diffraction pattern reflects the atomic arrangement of the diffracting material, as depicted in Figure 1, serving as a fingerprint for the crystals.

Traditional methods involve a search-match process (Altomare et al., 2008) among numerous known powder diffraction patterns. Given a target XRD pattern, it iterates through candidates in databases until satisfactory alignment is achieved, known as the search-match method. Although this approach is prevalent, it is time-consuming (Altomare et al., 2008; Lv et al., 2024; International Centre for Diffraction Data) due to two issues: (1) high dependency on human intervention. Matching processes rely on domain-specific programs that require frequent human-assisted tuning during program execu-

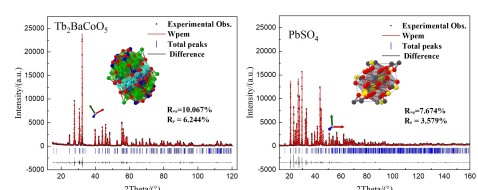

Figure 1: The X-ray diffraction patterns of crystal $Tb_2BaCoO_5$ and $PbSO_4$.

---

*Equal contribution.
†Corresponding authors (jialee@ust.hk, mezhangt@hkust-gz.edu.cn)

tion; (2) diffraction is a complex, multi-physical coupling process, making structure analysis challenging even for experienced domain experts.

Inspired by the remarkable progress of machine learning models, recent studies (Lee et al., 2020; Szymanski et al., 2021; Maffettone et al., 2021; Wang et al., 2020; Salgado et al., 2023) have attempted to train neural networks on simulated XRD datasets to increase the efficiency of symmetry identification. They treat XRD patterns as sequences and aim to classify them into specific crystal systems or space groups. More recent studies (Guo et al., 2024a;b; Li et al., 2024; Cao et al., 2025; Riesel et al., 2024; Lai et al., 2024) on powder XRD for crystal prediction and generation have employed advanced ML architectures, achieving significant progress and noteworthy results.

Despite significant progress, several limitations exist in current model development and evaluation:

- **Lack of a large-scale and high-quality dataset**: Previous research has largely been confined to specific materials (Fatimah et al., 2022; Lee et al., 2020; Wang et al., 2020). For example, Lee et al. (Lee et al., 2020) focus on mixtures of 38 distinct binary and ternary crystals in the Sr-Li-Al-O inorganic compounds. Additionally, the geometric similarity of crystals and experimental physical factors can result in XRD patterns with similar peak distributions. The introduction of large-scale databases presents new challenges. Therefore, sufficient coverage of structures and close alignment with experimental patterns under different physical conditions is necessary for developing models with high generalization ability.

- **Insufficient evaluation on real XRD patterns**: While symmetry classification aims to recognize symmetry from experimental measurements, most studies train and evaluate model performance on simulated data. Considering the potential bias between simulated and real XRD patterns, models may overfit simulated data and fail to generalize to real data.

- **Limited exploration on out-library identification**: Most studies focus on in-library identification. This involves training a neural network to generalize across different experimental settings for the same set of crystals. In contrast, out-library identification aims to generalize to entirely new crystals. This approach is crucial for discovering novel materials but receives insufficient attention.

The limited scale of experimental data presents two significant challenges in this field: (1) insufficient data to adequately test generative capabilities, and (2) difficulty in effectively utilizing such small-scale experimental datasets for model tuning, even within a transfer learning framework, let alone for training a model solely based on experimental data. To address this dilemma, we developed high-fidelity simulation data as a solution. Our results demonstrate no observable differences in model performance between simulated and experimental test datasets, providing evidence of the high quality of the simulations. Crafting a high-fidelity simulated XRD pattern dataset is challenging due to the complexity of domain-specific processes, such as determining crystal stability, configuring instruments, and accounting for physical environmental factors. In this work, we present a novel XRD simulation method (Appendix B.4) that incorporates comprehensive physical interactions. Using this method, we developed SimXRD, the largest open-source and physically detailed dataset aimed at advancing this interdisciplinary field. We source crystals from the Materials Project (MP) (Jain et al., 2013) database and apply rigorous filtering to remove crystals with broken symmetry, duplicates, or discrepancies in space group classification. Consequently, SimXRD comprises 4,065,346 X-ray powder diffraction patterns covering 119,569 distinct crystal structures simulated under various conditions, including grain size, orientation, internal stress, inelastic scattering, thermal vibration, instrumental zero shift, noise and others.

We benchmark both the in-library and out-library symmetry identification performance on SimXRD of 21 models, which can be divided into three types of sequence models: convolution neural networks (i.e., the backbone of existing symmetry classification models), recurrent models, and transformers. Through extensive experiments, we discover that:

- **Long-tail distribution**: The class labels follow a heavy long-tailed distribution. Most models are biased toward the high-frequency classes and fail to predict the low-frequency classes, especially in space group classification. To provide insights into model designs, we evaluate the model performance under different objective functions and find that label smoothing and focal loss yield better results.

Table 1: Summaries of existing powder XRD datasets. ICSD refers to the commercial Inorganic Crystal Structure Database. MP denotes the open-sourced Material Project.

| Dataset | #XRD Pattern | #Structure | Open Access | Simulated | Crystal Source | Year |
|---|---|---|---|---|---|---|
| RRUFF (Lafuente et al., 2015) | 3,002 | 3,002 | ✓ | × | - | 2015 |
| XRDSP (Suzuki et al., 2020) | 169,536 | 169,536 | ✓ | ✓ | ICSD | 2020 |
| CNN (Lee et al., 2020) | 1,785,405 | 170 | × | ✓ | ICSD | 2020 |
| PQNet (Dong et al., 2021) | 250,000 | 1 | ✓ | ✓ | ICSD | 2021 |
| XRDAutoAnalyzer (Szymanski et al., 2021) | 38,250 | 150 | ✓ | ✓ | ICSD | 2021 |
| XRDIsAllYouNeed (Lee et al., 2022) | 328,503 | 189,476&139,027[1] | × | ✓ | ICSD&MP | 2022 |
| AdvancedXRDAnalysis (Lee et al., 2023) | 29,569,650 | 197,131 | × | ✓ | ICSD | 2023 |
| CrySTINet (Chen et al., 2024) | 100 | 100 | ✓ | ✓ | ICSD | 2024 |
| CPICANN (Cao, 2024) | 692,190 | 23,073 | ✓ | ✓ | COD | 2024 |
| **SimXRD** | **4,065,346** | **119,569** | ✓ | ✓ | **MP** | **2024** |

- **Out-library generalization**: Out-library classification is challenging, as it requires models to learn from long-tailed data and generalize to XRD patterns of unobserved crystals. Thus, the model performance of out-library classification is generally reduced compared to that of in-library classification.

- **Experimental data generalization**: The ability to generalize to real experimental data is a gold standard for symmetry identification models. Remarkably, we found that models trained on the proposed SimXRD are able to achieve comparable or even better results on the real dataset (RRUFF), validating the contribution of our dataset construction.

We highlight that our findings provide the research community a clearer insight into the current progress in XRD analysis. Additionally, we have made the SimXRD database, simulation code, benchmark models, evaluation process and tutorial notebooks into a repository: `https://github.com/Bin-Cao/SimXRD`.

## 2 RELATED WORK

### 2.1 EXISTING DATASETS

Several datasets have been developed to train neural models for crystallography. Table 1 provides a summary of these datasets, compared across several key dimensions:

- **Dataset Size**: Existing open-source databases either contain relatively small structures tailored to specific purposes (Chen et al., 2024; Dong et al., 2021; Szymanski et al., 2021; Lee et al., 2020) or consider limited environmental settings (Chen et al., 2024; Suzuki et al., 2020). Crystal structure databases form the backbone of XRD pattern databases. Although the database developed by Lee et al. (Lee et al., 2022; 2023) encompasses nearly all the crystals in the Inorganic Crystal Structure Database (ICSD) (Zagorac et al., 2019) and the Materials Project (MP) (Jain et al., 2013), it contains a significant number of repeated crystals with broken symmetries. Additionally, it does not sufficiently account for various physical conditions, resulting in a relatively small XRD pattern database. SimXRD, by contrast, employs the entire MP database of January 2024, totaling 154,718 crystal structures. During the generation of the SimXRD database, we applied screening methods to improve crystal quality. Additionally, we simulate XRD patterns under various environments to meet the academic-industry requirements.

- **Availability**: Most pattern datasets are openly available, as indicated in Table 1, though some datasets can only be retrieved upon reasonable request (Lee et al., 2020; 2022; 2023). Additionally, published diffraction pattern datasets are saved in various formats, requiring specific data loaders provided by the authors. In contrast, the SimXRD-4M dataset developed in our work is fully accessible and features an easier workflow for machine learning training, easily integrable with TensorFlow or PyTorch frameworks.

- **Simulation vs. Experiments**: Among the existing datasets, only the RRUFF Project (Lafuente et al., 2015) aims to create a comprehensive set of high-quality spectral data from well-characterized minerals. Since its inception in 2005 (RRUFF, 2005), RRUFF has compiled a collection of 3,002 high-quality experimental powder XRD patterns as of March 2024. Conducting

---

[1]Union of ICSD and MP database, existing the same structures.

the experiments is time-consuming. Thus, recent studies have employed domain-specific software to generate simulated data. Earlier studies primarily relied on widely-used software tools, such as Pymatgen (Ong et al., 2013), FullProf (Rodríguez-Carvajal, 2001), and GSAS-II (Toby & Von Dreele, 2013), to simulate patterns for training. However, these approaches often showed varying degrees of performance degradation when applied to experimental test sets. SimXRD addresses these limitations by employing the newly developed PysimXRD, which accounts for a wider range of real-world conditions, including grain size, internal stress, external temperature variations, grain orientation, instrument drift, instrument noise, detector geometry, and scattering-induced background. This comprehensive approach ensures higher fidelity in data generation, improving the robustness of the resulting models.

## 2.2 SYMMETRY IDENTIFICATION

Most existing methods for symmetry identification rely on one-dimensional convolutional neural networks for building classification models. The models developed in recent years have primarily focused on CNN architectures (Dong et al., 2021; Wang et al., 2020; Park et al., 2017). In particular, they focus on designing CNNs with deep layers (Lee et al., 2022) and large kernel sizes (Le et al., 2023), ensemble CNNs (Maffettone et al., 2021; Szymanski et al., 2021), and CNNs without dropout (Lee et al., 2020) and pooling layers (Salgado et al., 2023). However, whether CNNs can address long-tailed distributions, as well as the performance of other sequence models, remain underexplored.

## 3 SIMXRD-4M DATASET

### 3.1 PRELIMINARIES

**Crystal symmetry** Symmetry in crystallography is a fundamental property of the orderly arrangements of atoms in crystals (Su et al., 2024). Each atomic arrangement possesses certain symmetry elements, which are transformations that leave the arrangement unchanged. These elements include rotation, translation, reflection, and inversion. The specific symmetry elements present in a crystalline solid determine its shape and influence its physical properties. Crystals are classified based on their geometry and symmetry elements, falling into one of 7 *crystal systems*, and could further divide into 230 *space groups* (Clegg, 2023). Further details are provided in AppendixB.1.

**XRD pattern** Powder XRD patterns offer a one-dimensional representation of the three-dimensional diffraction pattern and are the most common experimental measurement. Specifically, each one-dimensional powder X-ray diffraction pattern in SimXRD is presented in the format of d-I (lattice plane distance-intensity), where the x-axis represents lattice plane distances and the y-axis represents corresponding intensities, as shown in Figure 2. The diffraction pattern we provide is a 1D tensor with 3501 elements. These points represent lattice plane distances, equally spaced from 1.199 Å to 8.853 Å, with a step size of 0.0176 Å. This corresponds to a typical experimental setup for a two-theta diffractometer (copper target), where the diffraction angle is scanned from $10°$ to $80°$ with a step size of $0.02°$. The d-I pattern is independent of the wavelength of the incident X-rays, making it universally applicable across all experimental X-ray sources. Lattice plane distances denote the detected atomic directions. Some distances lack peaks due to geometric constraints from the lattice cell and extinction effects (Lund et al., 2010) under diffraction. The relative intensity is normalized, with the maximum intensity set to 100, making the pattern independent of the incident X-ray intensity.

### 3.2 DATA GENERATION AND PROCESSING

**Crystal data** The crystal structures used in this study were sourced from Material Project (Jain et al., 2013), a comprehensive, searchable database containing information on solid-state materials and molecules. It provides detailed data on the physical properties of these materials, such as elastic tensors, band structures, and formation energies, derived from electronic structure calculations, providing a more comprehensive reference for crystal studies. We utilize the latest dataset, denoted as MP-2024.1, which includes a total of 154,718 crystallographic structures as of January 2024.

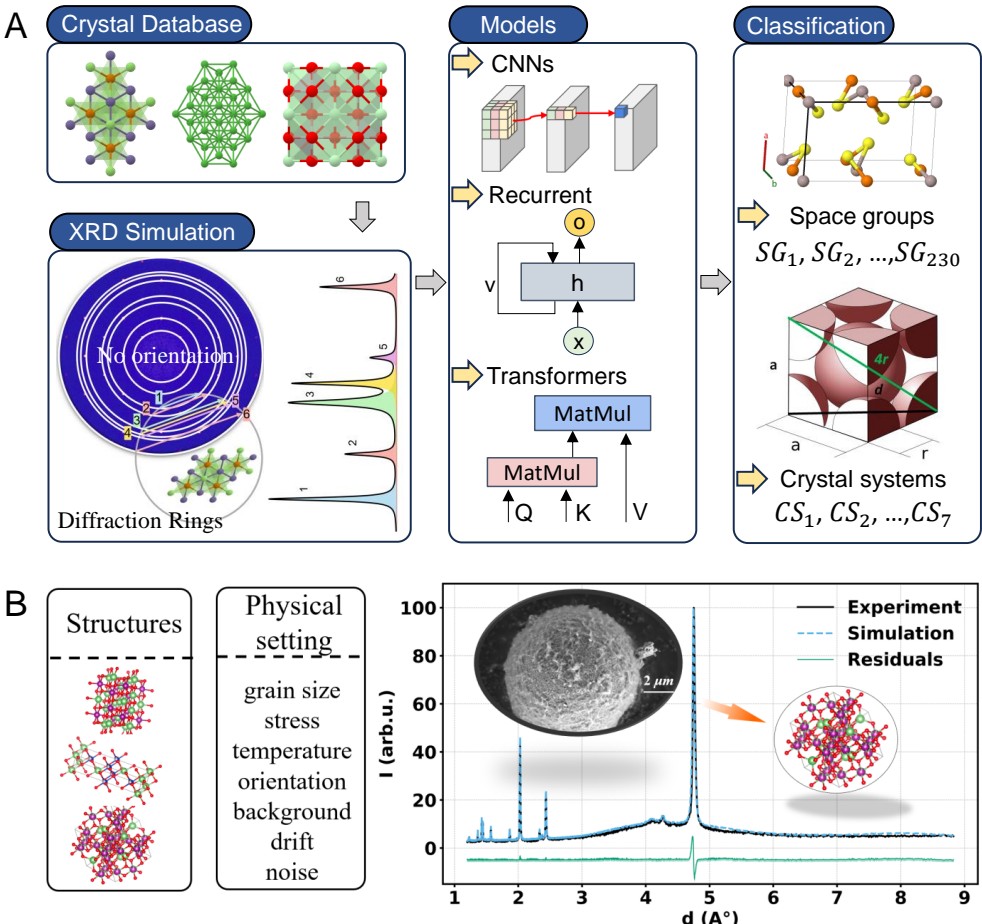

Figure 2: (A) The workflow of the large simulated XRD database and symmetry classification benchmark consists of four main components: (1) Retrieval of crystal data from the Materials Project database. (2) Multiphysical coupling simulations to generate high-fidelity powder XRD patterns. (3) Application of three machine learning model types, CNNs, recurrent models, and transformers, for benchmarking the database.(4) Symmetry classification, which maps powder XRD patterns to specific space groups and crystal systems. (B) Experimental XRD pattern of a Li-rich layered oxide cathode ($Li_2MnO_3$) was compared with simulated pattern generated using PysimXRD. The simulation incorporates multiphysical coupling, producing patterns that closely match experimental measurement with minimal residual errors.

**Crystal Filtering**    To ensure consistency between recorded space group numbers (crystal systems) and atomic arrangements, and to enhance the quality of crystal structures, we performed a thorough examination of each structure in the MP database. We used Spglib (Togo et al., 2024), a library designed for identifying and managing crystal symmetries, to analyze all structures. Structures exhibiting broken symmetry, duplication, or discrepancies in space groups were excluded. Additionally, only structures containing up to 500 atoms per lattice cell (conventional unit cell) were retained, effectively covering nearly all inorganic materials in the MP dataset. A total of 119,569 crystal structures were screened. Further details are given in Appendix A.1.

**Simulation in SimXRD**    Figure 2 illustrates the overall simulation process. The experimental XRD patterns depend on both the crystal's intricate structure and practical factors, including the sample's state and instrumental parameters. Our goal is to simulate this process using domain-specific custom software Pysimxrd (Appendix B.4). First, we specify the environmental settings and then compute the XRD patterns for the given crystals. By varying the environmental param-

eters, we generate all samples in the SimXRD dataset. We systematically analyze the physical conditions affecting the XRD profiles, categorizing them into factors that influence the sample's internal structure and those affect instrumental measurements. These parameters, along with their respective value ranges, are outlined in the Appendix B.4. For each input crystal, we randomly combine these physical conditions within reasonable ranges, repeating the process 33 times to simulate XRD patterns across diverse environments. Each XRD pattern is standardized, with the x-axis values (lattice plane distance) uniformly set based on the experimental settings, resulting in a vector of 3501 dimensions. The y-axis values (diffraction intensity) are simulated according to the crystal structure and diffraction conditions.

## 3.3 DATA ANALYSIS

**Long-tailed distribution**   The distribution of space groups on a logarithmic scale is shown in Figure 3. The space groups in the MP dataset are ranked in descending order, and their frequencies are plotted on a logarithmic scale. This reveals differences in frequency spanning several orders of magnitude. The data clearly exhibit a highly imbalanced, long-tailed distribution. Since MP-2024.1 serves as a comprehensive database of commonly occurring material systems, this distribution imbalance is inherent to the physical population of crystals in nature. This imbalance can lead to biased model performance, as the model may be more inclined to predict the majority class, leading to lower accuracy for the minority classes. Since minority crystals are also important for material discovery, addressing long-tailed sequence learning is crucial for symmetry identification. However, existing models do not consider this challenge, making it necessary to evaluate their performance across different classes. The detailed space group distribution and the energy distribution of crystals are presented in Appendix A.1.

**Case study**   A case study is conducted to investigate the difference between experimental and simulated XRD patterns of $Li_2MnO_3$. Initially, the physical states of the tested sample and the experimental settings were obtained using a refinement method (see Appendix B.7). Subsequently, the XRD pattern was generated inversely using the Pysimxrd. Figure 2 (B) shows the results where we can observe that the generated profile exhibits extreme similarity to the experimental measurements(Lei et al., 2024). Therefore, by altering the physical states within a reasonable range, other patterns corresponding to different experimental settings can be generated. Such simulation allows us to obtain high-quality data in a relatively efficient manner, compared to experimental patterns.

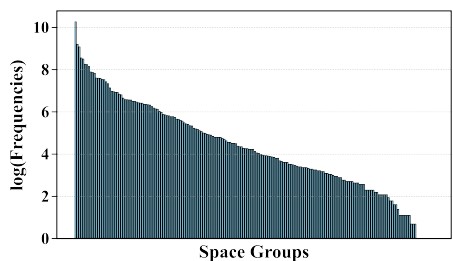

Figure 3: The distribution of space groups on logarithmic scale

## 4 BENCHMARK

### 4.1 PROBLEM DEFINITION

**Symmetry classification**   Formally, SimXRD treats the problem as a multi-class sequence classification task. Given the XRD pattern $\boldsymbol{X} = [x_1, x_2, \cdots, x_n] \in \mathbb{R}^n$ where $x_i$ is the $i$-th intensity value and $\boldsymbol{X}$ is arranged by lattice plane distance, the goal is to predict its symmetry $\boldsymbol{Y} \in \mathbb{R}^k$. Here $n$ is the feature dimension which is 3501 in our dataset. For crystal system classification, $k = 7$, and for space group classification, $k = 230$. We did not consider practical factors such as average grain size and stress as input variables, since these are not directly measurable in real-world diffraction experiments.

**In/Out-library crystallographic tasks**   In-library classification aims to generalize across different experimental environments for the same set of structures. Consequently, the training and testing datasets consist the same structures but with varying simulation parameters. In contrast, out-library classification seeks to generalize to different crystals. Therefore, the training and testing datasets consist of different types of crystals. More details are provided in Appendix B.2.

Table 2: Results of in library crystal system classification and space group classification. Inference time is measured for a batch size of 100 samples and experiments are run on a GeForce RTX 3090 GPU.

| Model | # Conv. | # Dropout | # Pooling | Ensemble | Ref. | Crystal System | | | | | Space Group | | | | |
|---|---|---|---|---|---|---|---|---|---|---|---|---|---|---|---|
| | | | | | | Accuracy | F1 | Precision | Recall | Time (ms) | Accuracy | F1 | Precision | Recall | Time (ms) |
| CNN1 | 3 | ✓ | AvgPool | × | (Park et al., 2017) | 0.559 | 0.418 | 0.427 | 0.431 | 3.2 | 0.241 | 0.002 | 0.001 | 0.005 | 3.4 |
| CNN2 | 2 | × | MaxPool | × | (Lee et al., 2020) | 0.466 | 0.236 | 0.222 | 0.283 | 1.8 | 0.241 | 0.002 | 0.001 | 0.005 | 1.9 |
| CNN3 | 3 | × | MaxPool | × | (Lee et al., 2020) | 0.531 | 0.328 | 0.315 | 0.369 | 3.5 | 0.241 | 0.002 | 0.001 | 0.005 | 3.6 |
| CNN4 | 7 | ✓ | MaxPool | × | (Wang et al., 2020) | 0.316 | 0.069 | 0.045 | 0.143 | 1.4 | 0.241 | 0.002 | 0.001 | 0.005 | 1.4 |
| CNN5 | 3 | ✓ | AvgPool | ✓ | (Maffettone et al., 2021) | 0.517 | 0.394 | 0.489 | 0.378 | 0.6 | 0.295 | 0.022 | 0.025 | 0.026 | 0.6 |
| CNN6 | 7 | ✓ | MaxPool | × | (Dong et al., 2021) | 0.316 | 0.069 | 0.045 | 0.143 | 65.0 | 0.241 | 0.002 | 0.001 | 0.005 | 65.0 |
| CNN7 | 6 | ✓ | MaxPool | ✓ | (Szymanski et al., 2021) | 0.862 | 0.863 | 0.887 | 0.845 | 6.8 | 0.588 | 0.124 | 0.147 | 0.130 | 6.9 |
| CNN8 | 14 | ✓ | MaxPool | × | (Lee et al., 2022) | 0.377 | 0.162 | 0.148 | 0.221 | 3.1 | 0.241 | 0.002 | 0.001 | 0.005 | 3.1 |
| CNN9 | 3 | ✓ | MaxPool | × | (Le et al., 2023) | 0.795 | 0.817 | 0.826 | 0.810 | 1.9 | 0.599 | 0.597 | 0.681 | 0.556 | 1.9 |
| CNN10 | 4 | ✓ | MaxPool | × | (Le et al., 2023) | 0.870 | 0.888 | 0.892 | 0.885 | 1.8 | 0.705 | 0.792 | **0.853** | 0.759 | 1.8 |
| CNN11 | 3 | ✓ | None | × | (Salgado et al., 2023) | **0.902** | **0.922** | **0.932** | **0.914** | 4.8 | **0.758** | 0.750 | 0.735 | **0.828** | 4.8 |
| MLP | | | | | | 0.316 | 0.069 | 0.045 | 0.143 | 1.6 | 0.241 | 0.002 | 0.001 | 0.005 | 1.6 |
| RNN | | | | | | 0.381 | 0.183 | 0.200 | 0.202 | 8.0 | 0.245 | 0.003 | 0.002 | 0.007 | 8.1 |
| LSTM | | | | | | 0.728 | 0.743 | 0.762 | 0.728 | 14.6 | 0.515 | 0.156 | 0.224 | 0.151 | 14.6 |
| GRU | | | | | | 0.765 | 0.788 | 0.802 | 0.777 | 15.1 | 0.575 | 0.273 | 0.400 | 0.251 | 15.1 |
| Bidirectional-RNN | | | | | | 0.365 | 0.155 | 0.197 | 0.185 | 14.7 | 0.245 | 0.003 | 0.002 | 0.007 | 14.7 |
| Bidirectional-LSTM | | | | | | 0.791 | 0.814 | 0.825 | 0.805 | 29.1 | 0.559 | 0.384 | 0.533 | 0.346 | 29.1 |
| Bidirectional-GRU | | | | | | 0.800 | 0.826 | 0.840 | 0.816 | 30.3 | 0.627 | 0.451 | 0.609 | 0.408 | 30.3 |
| Transformer | | | | | | 0.338 | 0.127 | 0.172 | 0.155 | 83.4 | 0.241 | 0.002 | 0.001 | 0.005 | 83.5 |
| iTransformer | | | | | | 0.627 | 0.611 | 0.652 | 0.599 | 1.9 | 0.388 | 0.135 | 0.320 | 0.118 | 1.9 |
| PatchTST | | | | | | 0.720 | 0.752 | 0.766 | 0.740 | 3.3 | 0.631 | **0.811** | 0.850 | 0.784 | 3.3 |

## 4.2 Experimental Setting

**Dataset split**  For in-library classification, a fundamental task in crystallography, the dataset is randomly split according to the types of simulated environments, resulting in 119,569 × 30 training instances, 119,569 × 1 validation instances, and 119,569 × 2 testing instances. The data recorded for training, validation, and testing include crystals from different simulated environments (e.g., grain size, stress, temperature). Under out-of-library settings, the training and testing XRD patterns are generated from non-overlapping crystals. This setup yields 83,698 × 33 training instances, 11,957 × 33 validation instances, and 23,914 × 33 testing instances.

**Baselines**  We consider the following three types of sequence classification models:

- **CNN-based Models**: We evaluate all existing CNN architectures proposed for symmetry identification. Since most models do not have specific names, we classify them based on their number of convolution layers, pooling layers, and whether they use ensemble learning or dropout layers. Table 2 summarizes the existing 11 CNN models.

- **Recurrent Models**: We select three basic recurrent neural networks: RNN (Medsker & Jain, 1999), LSTM (Hochreiter & Schmidhuber, 1997), and GRU (Chung et al., 2014). Additionally, since XRD patterns can be viewed from both forward and backward, we also evaluate the performance of bidirectional RNN, LSTM, and GRU (Schuster & Paliwal, 1997).

- **Transformers**: Transformers have proven effective in variety of sequence modeling tasks. We evaluate the performance of the raw transformer (Vaswani et al., 2017) and two advanced transformer models - iTransformer (Liu et al., 2023) and PatchTST (Nie et al., 2023).

Besides these models, we also include MLP as a straightforward baseline. For recurrent models and transformers, we first use them to learn sequence representations, and then employ an MLP to predict the symmetry.

**Implementation details**  We use the following hyper-parameters across all experiments: Batch size of 128 and learning rate of $2.5 \times 10^{-4}$. All models are trained for 50 epochs with an early stopping patience of 3. We use the Cross-Entropy function to measure the difference between predictions and the ground truth. The performance of the models is evaluated using accuracy, macro F1-score, macro precision, and macro recall as metrics. All models are implemented using the PyTorch (Paszke et al., 2019) library and trained on GeForce RTX 3090 GPU.

Table 3: Results of weighted classification, label smoothing, and Focal loss on crystal system and space group classification.

| | Weighted classification | | | Label smoothing | | | Focal loss | | |
|---|---|---|---|---|---|---|---|---|---|
| **Crystal System Classification** | | | | | | | | | |
| Training | Weighted classification | | | Label smoothing | | | Focal loss | | |
| Metric | F1 | Precision | Recall | F1 | Precision | Recall | F1 | Precision | Recall |
| CNN1 | 0.565 | 0.538 | 0.642 | 0.578 | 0.587 | 0.575 | 0.607 | 0.608 | 0.610 |
| CNN2 | 0.690 | 0.659 | 0.734 | 0.497 | 0.493 | 0.506 | 0.511 | 0.506 | 0.518 |
| CNN3 | 0.699 | 0.665 | 0.765 | 0.723 | 0.765 | 0.698 | 0.536 | 0.529 | 0.546 |
| CNN4 | 0.229 | 0.191 | 0.390 | 0.066 | 0.043 | 0.142 | 0.066 | 0.043 | 0.142 |
| CNN5 | 0.397 | 0.379 | 0.468 | 0.382 | 0.431 | 0.375 | 0.313 | 0.339 | 0.321 |
| CNN6 | 0.069 | 0.046 | 0.142 | 0.069 | 0.046 | 0.142 | 0.069 | 0.046 | 0.142 |
| CNN7 | 0.740 | 0.710 | 0.793 | 0.863 | 0.887 | 0.844 | 0.851 | 0.871 | 0.836 |
| CNN8 | 0.546 | 0.533 | 0.625 | 0.566 | 0.590 | 0.559 | 0.554 | 0.687 | 0.547 |
| CNN9 | 0.780 | 0.748 | 0.824 | 0.802 | 0.820 | 0.788 | 0.814 | 0.803 | 0.807 |
| CNN10 | 0.800 | 0.767 | **0.848** | 0.877 | 0.898 | 0.860 | 0.844 | 0.857 | 0.834 |
| CNN11 | **0.811** | **0.797** | 0.845 | **0.931** | **0.943** | **0.921** | **0.887** | **0.895** | **0.882** |
| MLP | 0.069 | 0.046 | 0.142 | 0.069 | 0.046 | 0.142 | 0.069 | 0.046 | 0.142 |
| RNN | 0.154 | 0.184 | 0.236 | 0.180 | 0.200 | 0.196 | 0.167 | 0.197 | 0.187 |
| LSTM | 0.086 | 0.153 | 0.171 | 0.706 | 0.721 | 0.697 | 0.701 | 0.712 | 0.701 |
| GRU | 0.769 | 0.737 | 0.821 | 0.783 | 0.799 | 0.771 | 0.727 | 0.742 | 0.717 |
| Bidirectional-RNN | 0.169 | 0.184 | 0.238 | 0.144 | 0.192 | 0.176 | 0.159 | 0.150 | 0.179 |
| Bidirectional-LSTM | 0.717 | 0.684 | 0.789 | 0.795 | 0.819 | 0.775 | 0.622 | 0.653 | 0.602 |
| Bidirectional-GRU | 0.704 | 0.674 | 0.777 | 0.861 | 0.863 | 0.863 | 0.845 | 0.843 | 0.847 |
| Transformer | 0.134 | 0.161 | 0.178 | 0.130 | 0.169 | 0.156 | 0.129 | 0.171 | 0.156 |
| iTransformer | 0.545 | 0.516 | 0.643 | 0.590 | 0.618 | 0.580 | 0.604 | 0.630 | 0.598 |
| PatchTST | 0.672 | 0.643 | 0.745 | 0.710 | 0.735 | 0.691 | 0.714 | 0.722 | 0.709 |
| **Space Group Classification** | | | | | | | | | |
| CNN1 | 0.000 | 0.000 | 0.005 | 0.002 | 0.001 | 0.005 | 0.002 | 0.001 | 0.005 |
| CNN2 | 0.000 | 0.000 | 0.005 | 0.002 | 0.001 | 0.005 | 0.002 | 0.001 | 0.005 |
| CNN3 | 0.000 | 0.000 | 0.005 | 0.002 | 0.001 | 0.005 | 0.002 | 0.001 | 0.005 |
| CNN4 | 0.001 | 0.000 | 0.005 | 0.002 | 0.001 | 0.005 | 0.002 | 0.001 | 0.005 |
| CNN5 | 0.033 | 0.028 | 0.056 | 0.002 | 0.001 | 0.005 | 0.002 | 0.001 | 0.005 |
| CNN6 | 0.000 | 0.001 | 0.005 | 0.002 | 0.001 | 0.005 | 0.002 | 0.001 | 0.005 |
| CNN7 | 0.220 | 0.217 | 0.377 | 0.124 | 0.143 | 0.131 | 0.250 | 0.225 | 0.217 |
| CNN8 | 0.000 | 0.000 | 0.005 | 0.002 | 0.001 | 0.005 | 0.002 | 0.001 | 0.005 |
| CNN9 | **0.391** | **0.325** | **0.739** | 0.370 | 0.648 | 0.303 | 0.582 | 0.674 | 0.535 |
| CNN10 | 0.200 | 0.223 | 0.649 | 0.542 | 0.806 | 0.452 | **0.774** | **0.851** | 0.732 |
| CNN11 | 0.000 | 0.000 | 0.005 | 0.615 | **0.836** | 0.525 | 0.751 | 0.741 | **0.815** |
| MLP | 0.000 | 0.000 | 0.005 | 0.002 | 0.001 | 0.005 | 0.002 | 0.001 | 0.005 |
| RNN | 0.000 | 0.002 | 0.006 | 0.003 | 0.001 | 0.006 | 0.003 | 0.001 | 0.005 |
| LSTM | 0.000 | 0.000 | 0.006 | 0.143 | 0.210 | 0.137 | 0.140 | 0.206 | 0.137 |
| GRU | 0.215 | 0.180 | 0.610 | 0.185 | 0.261 | 0.176 | 0.227 | 0.192 | 0.653 |
| Bidirectional-RNN | 0.004 | 0.006 | 0.010 | 0.003 | 0.002 | 0.006 | 0.004 | 0.005 | 0.009 |
| Bidirectional-LSTM | 0.148 | 0.135 | 0.518 | 0.177 | 0.278 | 0.166 | 0.177 | 0.275 | 0.159 |
| Bidirectional-GRU | 0.088 | 0.099 | 0.276 | 0.297 | 0.475 | 0.246 | 0.472 | 0.638 | 0.416 |
| Transformer | 0.001 | 0.001 | 0.006 | 0.002 | 0.002 | 0.005 | 0.002 | 0.002 | 0.005 |
| iTransformer | 0.083 | 0.096 | 0.387 | 0.092 | 0.172 | 0.090 | 0.173 | 0.365 | 0.149 |
| PatchTST | 0.121 | 0.123 | 0.386 | **0.677** | 0.801 | **0.609** | 0.719 | 0.808 | 0.665 |

## 4.3 RESULTS

**In-library classification** Table 2 displays the baseline performance and inference time for the classification of crystal systems and space groups. Based on the results, we make the following observations:

- Most existing CNN models are unsuitable for symmetry identification within the large scale of the SimXRD database. It is clear that the crystal system classification accuracies of several CNNs, e.g., CNN 4 and 6 are on par with that of MLP. As shown in the additional results in Appendix A.2, the reason for this is that their predictions are heavily biased by high frequency classes, making them unable to predict low-frequency classes effectively. This limitation is even more apparent in space group classification, where the accuracy of most CNNs (i.e., CNN 1-4, 6, and 8) is 0.241, indicating that their outputs are almost fixed to the most frequent space group.

- Convolutional neural networks without pooling have achieved the best performance in most tasks, consistent with the findings of previous work (Salgado et al., 2023). Since the peak maximum and the relative value among peaks in XRD patterns are closely related to the atom arrangement of crystals, employing pooling layers may lead to information loss, which can affect peak identification and result in lower accuracy.

Table 4: Results of out-library symmetry identification.

| Task | Crystal System Classification | | | | Space Group Classification | | | |
|---|---|---|---|---|---|---|---|---|
| Model | Accuracy | F1 | Precision | Recall | Accuracy | F1 | Precision | Recall |
| CNN1 | 0.627 | 0.437 | 0.565 | 0.472 | 0.285 | 0.003 | 0.002 | 0.006 |
| CNN2 | 0.606 | 0.419 | 0.428 | 0.425 | 0.285 | 0.003 | 0.002 | 0.006 |
| CNN3 | 0.659 | 0.501 | 0.496 | 0.507 | 0.285 | 0.003 | 0.002 | 0.006 |
| CNN4 | 0.378 | 0.078 | 0.054 | 0.142 | 0.285 | 0.003 | 0.002 | 0.006 |
| CNN5 | 0.495 | 0.278 | 0.285 | 0.283 | 0.285 | 0.003 | 0.002 | 0.006 |
| CNN6 | 0.378 | 0.078 | 0.054 | 0.142 | 0.285 | 0.003 | 0.002 | 0.006 |
| CNN7 | 0.673 | 0.607 | 0.633 | 0.608 | 0.429 | 0.053 | 0.050 | 0.072 |
| CNN8 | 0.612 | 0.452 | 0.448 | 0.497 | 0.286 | 0.003 | 0.001 | 0.006 |
| CNN9 | 0.675 | 0.632 | 0.629 | 0.644 | 0.439 | 0.099 | 0.137 | 0.113 |
| CNN10 | 0.692 | 0.659 | 0.650 | 0.674 | 0.430 | 0.107 | 0.168 | 0.121 |
| CNN11 | 0.702 | 0.672 | 0.659 | 0.690 | 0.481 | 0.136 | 0.167 | 0.150 |
| MLP | 0.378 | 0.078 | 0.054 | 0.142 | 0.285 | 0.003 | 0.002 | 0.006 |
| RNN | 0.409 | 0.162 | 0.149 | 0.178 | 0.274 | 0.003 | 0.002 | 0.009 |
| LSTM | 0.657 | 0.575 | 0.583 | 0.589 | 0.431 | 0.070 | 0.092 | 0.085 |
| GRU | 0.707 | 0.678 | 0.656 | 0.709 | 0.480 | 0.110 | 0.143 | 0.125 |
| Bidirectional-RNN | 0.403 | 0.157 | 0.146 | 0.174 | 0.295 | 0.004 | 0.003 | 0.008 |
| Bidirectional-LSTM | 0.704 | 0.663 | 0.654 | 0.678 | 0.349 | 0.035 | 0.044 | 0.051 |
| Bidirectional-GRU | 0.722 | 0.699 | 0.697 | 0.705 | 0.498 | 0.138 | 0.192 | 0.149 |
| Transformer | 0.376 | 0.138 | 0.231 | 0.158 | 0.285 | 0.003 | 0.005 | 0.006 |
| iTransformer | 0.606 | 0.495 | 0.519 | 0.526 | 0.367 | 0.053 | 0.085 | 0.064 |
| PatchTST | 0.656 | 0.616 | 0.612 | 0.627 | 0.375 | 0.067 | 0.093 | 0.082 |

- Bidirectional recurrent models consistently outperform their unidirectional counterparts. These results highlight an important physical attribute of XRD patterns : ensuring the consistent retrieval of crystal information through bidirectional pattern reading. This property is distinct from previous sequence tasks such as text and time-series classification, necessitates further model design considerations.

- The raw transformer encounters difficulties when dealing with the long-tailed sequence problem. However, compared to the raw transformer, we observe that PatchTST achieves a significant performance improvement, demonstrating that subsequence-level patches are beneficial for peak identification and further enhancing classification performance.

**Long-tailed objective functions** The long-tailed distribution significantly impedes the symmetry identification of XRD patterns. To address this issue, we further train all baseline models using the following techniques: (1)Weighted classification that reweights training instances based on the inverse of class frequency; (2) Label smoothing that applies cross-entropy with targets smoothed towards a uniform distribution; (3) Focal loss that focuses on hard-to-classify examples by downweighting the loss for easy examples. The results are presented in Table 3. These results show that weighted classification does not improve model performance. This lack of improvement is likely because increasing the importance of minority classes leads to a decrease in accuracy for the majority classes. In contrast, label smoothing and focal loss generally generally yield better results for both crystal system and space group classification, suggesting promising research directions for model designs.

**Out-library classification** Different splitting mechanisms are essential for preliminary experiments. Consequently, we conduct experiments using out-of-library settings, where the training and testing XRD patterns come from non-overlapping structures. To ensure that the training, validation, and test datasets consist of distinct crystals, we randomly partition the dataset by crystal type. The results are detailed in Table 4. Based on the findings, we have the following observations:

- Performance decrease compared to in-library classification. This decline is anticipated, as out-of-library classification requires models to generalize across different crystal types, which presents a greater challenge.

- The relative model performances align with the results from in-library identification, with Bidirectional-GRU and CNN11 achieving higher scores. However, there is a noticeable decline in out-of-library cases, as the models need to account for varying extinction effects across different space groups, which introduces significant challenges.

- Models such as CNN1-4, CNN6, CNN8, MLP, and Transformer exhibit poor performance in out-of-library identification, with accuracy scores around 24%, close to random prediction. This

Table 5: Generalization experiments on RRUFF dataset.

| Task | Crystal System Classification | | | | Space Group Classification | | | |
|---|---|---|---|---|---|---|---|---|
| Model | Accuracy | F1 | Precision | Recall | Accuracy | F1 | Precision | Recall |
| CNN1 | 0.623 | 0.542 | 0.563 | 0.539 | 0.241 | 0.004 | 0.002 | 0.010 |
| CNN2 | 0.425 | 0.210 | 0.178 | 0.143 | 0.241 | 0.004 | 0.002 | 0.010 |
| CNN3 | 0.506 | 0.313 | 0.397 | 0.332 | 0.241 | 0.004 | 0.002 | 0.010 |
| CNN4 | 0.297 | 0.065 | 0.042 | 0.143 | 0.241 | 0.004 | 0.002 | 0.010 |
| CNN5 | 0.530 | 0.402 | 0.511 | 0.384 | 0.302 | 0.043 | 0.045 | 0.051 |
| CNN6 | 0.297 | 0.065 | 0.042 | 0.143 | 0.241 | 0.004 | 0.002 | 0.010 |
| CNN7 | 0.868 | 0.877 | 0.899 | 0.860 | 0.577 | 0.202 | 0.205 | 0.222 |
| CNN8 | 0.297 | 0.065 | 0.042 | 0.143 | 0.241 | 0.004 | 0.002 | 0.010 |
| CNN9 | 0.792 | 0.819 | 0.832 | 0.809 | 0.615 | 0.500 | 0.555 | 0.486 |
| CNN10 | 0.859 | 0.885 | 0.893 | 0.879 | 0.696 | 0.685 | 0.741 | 0.669 |
| CNN11 | **0.893** | **0.920** | **0.930** | **0.914** | **0.766** | **0.700** | **0.754** | **0.687** |
| MLP | 0.325 | 0.070 | 0.046 | 0.143 | 0.241 | 0.004 | 0.002 | 0.010 |
| RNN | 0.377 | 0.165 | 0.203 | 0.185 | 0.245 | 0.006 | 0.004 | 0.013 |
| LSTM | 0.723 | 0.735 | 0.747 | 0.731 | 0.522 | 0.230 | 0.275 | 0.230 |
| GRU | 0.774 | 0.801 | 0.817 | 0.788 | 0.575 | 0.360 | 0.421 | 0.357 |
| Bidirectional-RNN | 0.365 | 0.152 | 0.190 | 0.184 | 0.245 | 0.005 | 0.003 | 0.013 |
| Bidirectional- LSTM | 0.788 | 0.813 | 0.820 | 0.808 | 0.553 | 0.367 | 0.419 | 0.354 |
| Bidirectional-GRU | 0.804 | 0.845 | 0.864 | 0.834 | 0.636 | 0.448 | 0.512 | 0.429 |
| Transformer | 0.354 | 0.138 | 0.219 | 0.165 | 0.241 | 0.004 | 0.002 | 0.010 |
| iTransformer | 0.629 | 0.612 | 0.658 | 0.607 | 0.392 | 0.143 | 0.156 | 0.151 |
| PatchTST | 0.723 | 0.764 | 0.785 | 0.750 | 0.635 | 0.696 | 0.748 | **0.687** |

underscores the need for further development of domain-specific models to effectively address the challenges of out-of-library identification.

**Experimental data generalization** A generalization experiment is conducted using the experimental RRUFF dataset. We train and validate all baseline models on SimXRD and then test their performance on RRUFF. The results are presented in Table 5. The findings indicate that the models generally achieve consistent performance on both SimXRD and RRUFF. Notably, CNN 11 consistently delivers the best accuracy, even surpassing its performance on SimXRD (e.g., CNN 11 in space group classification). Models that exhibited lower accuracy on SimXRD (e.g., CNN 1-6) also perform poorly on RRUFF.

## 5 CONCLUSION

In this paper, we introduce SimXRD, the largest open-source XRD pattern dataset for symmetry identification. Data analysis reveals that the symmetry labels follow a long-tailed distribution. We evaluate 21 models on two different splitting patterns (in-library and out-of-library) and find that most existing models struggle to accurately predict the symmetry of low-frequency classes, even when addressing for class imbalance. This limitation hinders their real-world applicability. Our results emphasize the importance of modeling long-tailed sequence classification and conducting comprehensive comparison to accurately assess the capabilities of various models. To promote further research and advancements in symmetry identification algorithms, we make both SimXRD and the simulation code available as open-source.

**Limitations** While SimXRD improves upon previous datasets in terms of sample size, crystal quality, and accessibility, the long-tailed and imbalanced label distribution remains an inherent limitation. The distribution of crystallographic symmetry arises through natural processes and is invariant to human intervention, making such biases difficult to address in any dataset. Potential solutions include developing class-imbalanced algorithms or incorporating additional information about the target crystals.

**Future works** We view SimXRD as an evolving project and are committed to its ongoing development. Future plans include: (1) Developing long-tailed sequence classification models through techniques such as data augmentation (Chu et al., 2020; Zheng et al., 2024) or enhancing features via graph inputs Liu et al. (2024a); (2) Exploring training approaches like multi-stage learning to address the imbalanced nature of XRD patterns; (3) Designing data augmentation strategies Liu et al. (2024b) for XRD patterns, while ensuring that methods like random swapping do not generate instances that violate underlying physical principles.

ACKNOWLEDGMENTS

This work is supported by the Guangzhou-HKUST(GZ) Joint Funding Program (Nos. 2023A03J0003 and 2023A03J0103) and is funded by National Natural Science Foundation of China Grant No. 72371217, the Guangzhou Industrial Informatic and Intelligence Key Laboratory No. 2024A03J0628, the Nansha Key Area Science and Technology Project No. 2023ZD003, and Project No. 2021JC02X191.

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

# A    ADDITIONAL RESULTS

## A.1    LONG-TAILED DISTRIBUTION AND FORMATION ENERGY

The distributions of space groups and crystal systems are shown in Figure 4A and B. The formation energy ($E_f$) of a configuration represents the energy required or released during its formation. Thus, the formation process can be either endothermic or exothermic. A more negative formation energy indicates greater structural and thermal stability Turnbull (1956). Figures 4C and D illustrate the formation energy per atom across 154,718 crystal structures in MP-2024.1 and 119,569 crystal structures in SimXRD. Figure 4C highlights the high-quality crystal data in MP-2024.1, with 90.33% of structures exhibiting negative formation energy. In comparison, the crystal data in SimXRD shows a negative formation energy rate of 92.18%, indicating that the excluded crystals are primarily unstable. This high-quality structural data forms the foundation of XRD pattern databases.

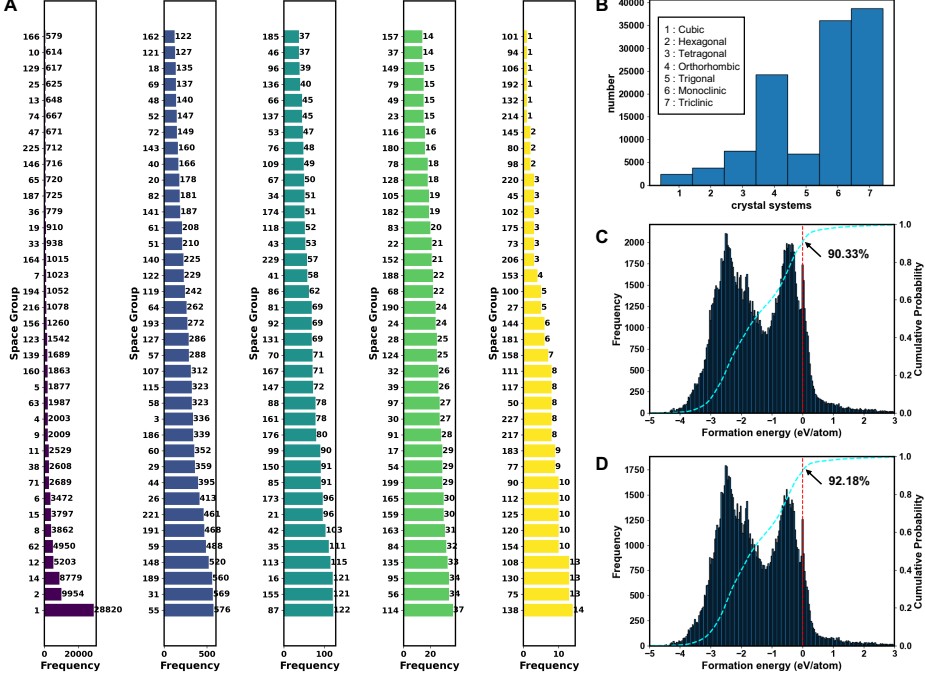

Figure 4: The statics of dataset. (A) The distribution of structures contained in the SimXRD database across various space groups (uniformly divided into 5 categories according to the frequency) and (B) crystal systems (C) The distribution of –formation energy/atom of crystals contained in MP-2024.1 and (D) SimXRD. Space groups 89, 103, and 104 have been excluded from Figure A to enhance clarity, each appearing only once.

## A.2    PERFORMANCE W.R.T. LONG-TAILED DISTRIBUTION

To further investigate the model's performance, we display the accuracy of each crystal system and space group at different frequencies in Figure 5 under in-library symmetry identification. To better visualize the model performance on space groups, we divided the results of 280 space groups into 5 categories according to their frequency (the same as Figure 4). From the results, we observe the following:

- In crystal system classification, the performance of most CNN models, the raw RNN, and transformers is affected by the frequency of crystal systems. They cannot predict the XRD patterns of Cubic and Hexagonal symmetry (their accuracies are almost zero). In contrast, multiple domain-specific CNN (i.e., CNN 7, 9, 10, 11), LSTM, GRU, and advanced Transformers (i.e., iTransformer and PatchTST) can mitigate long-tail distributions.

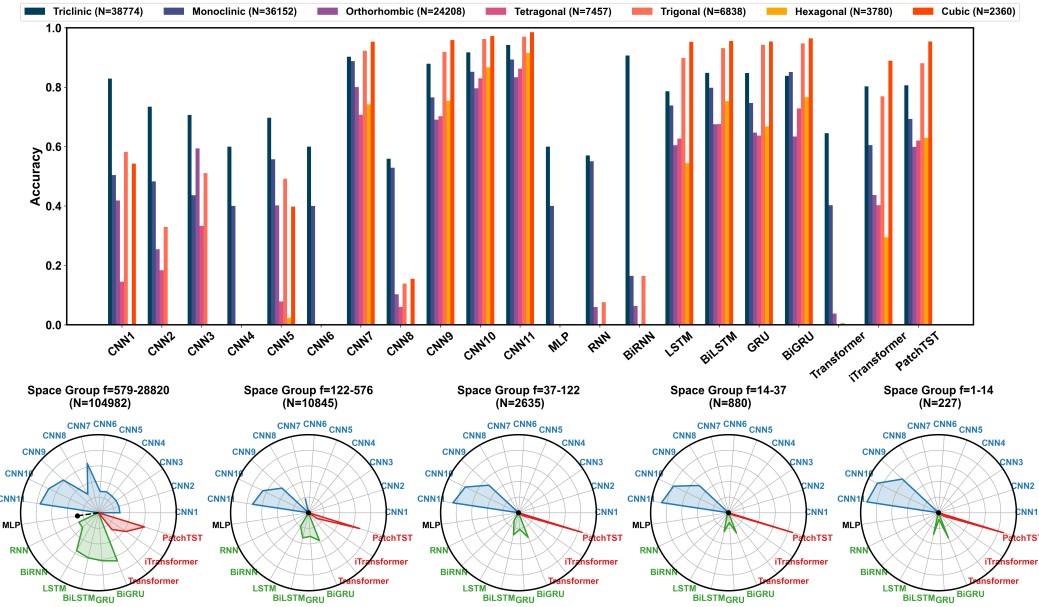

Figure 5: Classification accuracy of each model w.r.t. crystal systems and space groups. The same as Figure 4(A), Space groups are classified into 5 categories based on their frequency. $N$ denotes the number of crystals.

- Space group classification is more challenging than crystal system classification. Most models can only predict space groups with high frequency and struggle predict most low-frequency space groups. Surprisingly, PatchTST achieves comparable performance with the best models in all low-frequency space groups. Given that PatchTST is a model specifically tailored for time series forecasting, its performance on XRD pattern classification still has considerable room for improvement.

## A.3  PARAMETERS INTERVALS STUDY

To study how simulation parameters influence model performance on SimXRD, we perform an additional comparison by dividing the testing instances into three distinct intervals based on their key simulation parameters:

- **Category 1:** Grain size (nm) $\in [2, 10]$, atomic thermal offset (Å) $\in [0.4, 0.5]$, orientation randomness (%) $\in [30, 40]$, zero shifting (°) $\in [1, 1.2]$.

- **Category 2:** Grain size (nm) $\in [40, 50]$, atomic thermal offset (Å) $\in [0, 0.1]$, orientation randomness (%) $\in [0, 10]$, zero shifting (°) $\in [0, 0.2]$.

- **Category 3:** The parameter domain excludes Category 1 and Category 2.

The three parameter categories represent three simulated experimental scenarios: Category 1 reflects a high degree of peak broadening and external influence, while Category 2 is aligned with relative ideal conditions, exhibiting lower broadening and noise. Category 3 represents an intermediate state between the two. The accuracy of space group classification is displayed in Table 6. The results indicate that the baselines generally perform better in Category 2 and exhibit the lowest accuracy in Category 1, underscoring the challenges of modeling experimental XRD patterns in more complex physical situations. These insights highlight the need for developing a high-fidelity database that encompasses diverse crystals and covers a sufficient range of practical scenarios.

Table 6: Parameter Categories comparison of crystal systems & space group

| Task | Crystal System Classification | | | Space Group Classification | | |
|---|---|---|---|---|---|---|
| Metric | Category1 | Category2 | Category3 | Category1 | Category2 | Category3 |
| CNN1 | 0.523 | 0.553 | 0.554 | 0.250 | 0.233 | 0.241 |
| CNN2 | 0.452 | 0.467 | 0.464 | 0.250 | 0.233 | 0.241 |
| CNN3 | 0.318 | 0.384 | 0.307 | 0.250 | 0.233 | 0.241 |
| CNN4 | 0.375 | 0.305 | 0.302 | 0.250 | 0.233 | 0.241 |
| CNN5 | 0.500 | 0.545 | 0.526 | 0.312 | 0.286 | 0.295 |
| CNN6 | 0.375 | 0.305 | 0.302 | 0.250 | 0.233 | 0.241 |
| CNN7 | 0.781 | 0.894 | 0.862 | 0.437 | 0.566 | 0.575 |
| CNN8 | 0.375 | 0.305 | 0.302 | 0.250 | 0.233 | 0.241 |
| CNN9 | 0.781 | 0.823 | 0.785 | 0.343 | 0.646 | 0.609 |
| CNN10 | **0.812** | **0.908** | 0.868 | 0.406 | 0.727 | 0.704 |
| CNN11 | 0.656 | **0.908** | **0.895** | **0.562** | **0.823** | **0.775** |
| MLP | 0.312 | 0.327 | 0.324 | 0.250 | 0.233 | 0.241 |
| RNN | 0.468 | 0.383 | 0.367 | 0.218 | 0.236 | 0.245 |
| LSTM | 0.593 | 0.761 | 0.727 | 0.281 | 0.577 | 0.515 |
| GRU | 0.531 | 0.793 | 0.765 | 0.437 | 0.596 | 0.575 |
| Bidirectional-RNN | 0.343 | 0.362 | 0.365 | 0.250 | 0.236 | 0.245 |
| Bidirectional-LSTM | 0.781 | 0.837 | 0.790 | 0.437 | 0.600 | 0.559 |
| Bidirectional-GRU | 0.656 | 0.818 | 0.800 | 0.437 | 0.596 | 0.575 |
| Transformer | 0.375 | 0.364 | 0.338 | 0.250 | 0.233 | 0.240 |
| iTransformer | 0.500 | 0.662 | 0.627 | 0.312 | 0.378 | 0.388 |
| PatchTST | 0.562 | 0.745 | 0.720 | 0.437 | 0.672 | 0.631 |

## A.4 FEATURE ANALYSIS

We also analyze the relative feature importance map to provide model interpretability, as shown in Figure 6. By masking features in powder XRD patterns, we observe the impact of the masked features on the model's inference by calculating the relative accuracy drop. This approach reflects the relative importance of the masked features for model classification. As expected, masking critical features results in a significant drop in accuracy.

To further investigate, we categorize peak features into five groups based on their intensity: (1) Peaks with intensities within the 0–20% maximum intensity range, (2) Peaks within the 20–40% intensity range, (3) Peaks within the 40–60% intensity range, (4) Peaks within the 60–80% intensity range, and (5) Peaks within the 80–100% intensity range. We evaluate the model's inference performance with these masked features, as shown in Figure 6. The results demonstrate a clear trend: when high-intensity peaks, particularly those in category 5, are masked, a significant accuracy drop is observed across all baselines. This indicates that the ML model heavily relies on high-intensity peaks, similar to the "characteristic peaks" identified in search-match approaches.

Interestingly, for models that perform well on relative tasks, masking peaks in categories 3 or 4 also noticeably affects performance. This suggests that these models capture more comprehensive pattern characteristics by giving attention to relatively weak intensity peaks. Such behavior is reasonable for deep neural networks, as they extract more detailed information rather than relying solely on the 'three strongest diffraction peaks'. This leads to better performance.

Comparing the results of CNN11, BiGRU, and PatchTST, three models that performed relatively well in these symmetry identification tasks, we observe distinct strategies for decision-making. CNN11 places greater emphasis on characteristic peaks in category 5, indicating that its inference approach aligns closely with the traditional search-match method. PatchTST follows a similar trend, but the relative accuracy drop is more pronounced, suggesting that PatchTST captures a broader range of peak characteristics compared to CNN11. In contrast, BiGRU exhibits a gradient in the accuracy drop, indicating that it selectively focuses on relatively strong peaks while progressively reducing emphasis on weaker peaks based on their intensity. This strategy is intuitively reasonable, as weaker peaks are more susceptible to noise. These observations suggest that bidirectional architectures, such as BiGRU, may provide a promising solution for studying powder XRD data, as concluded in the main text.

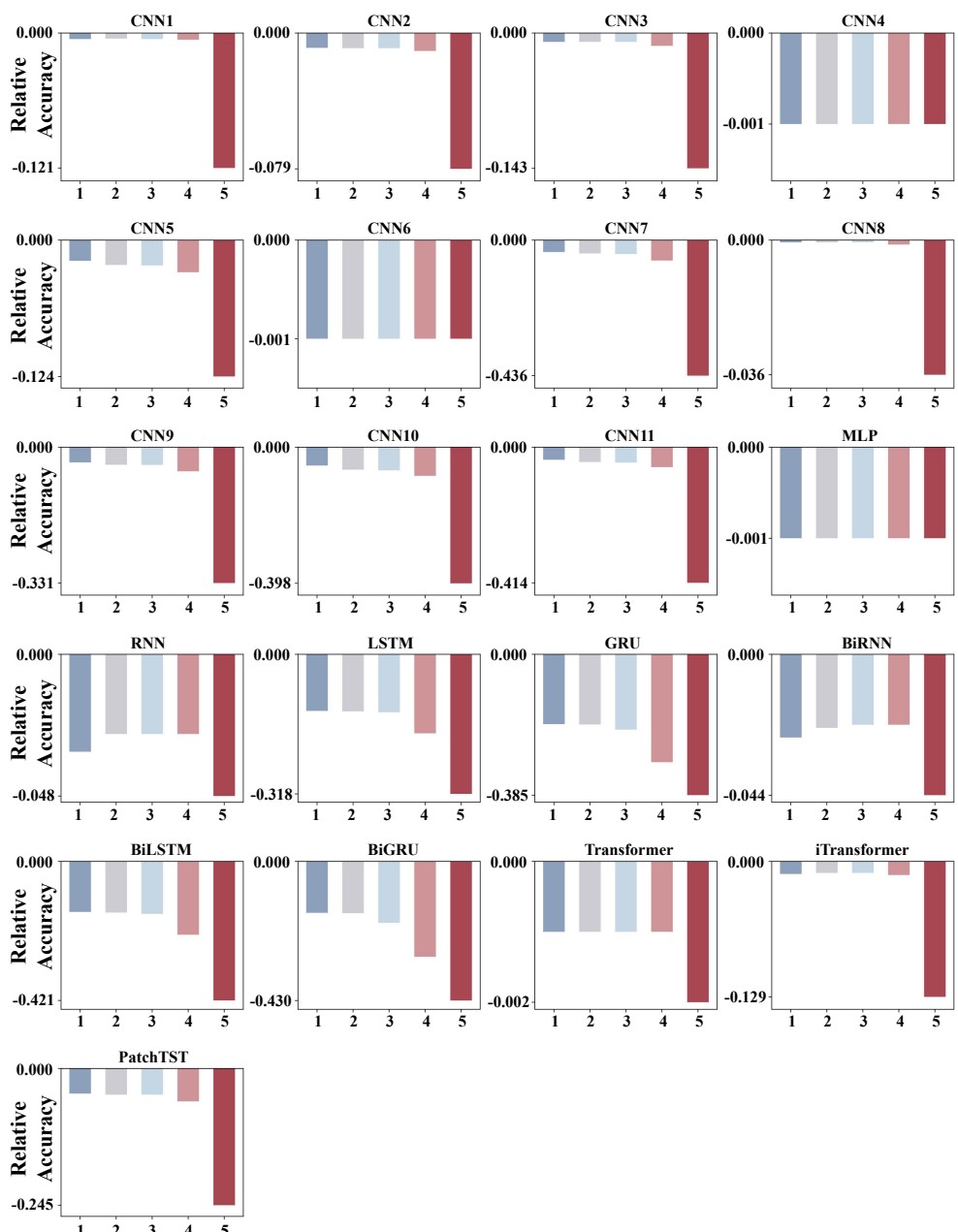

Figure 6: The relative classification accuracy drops when peak features are masked across all baselines. These features are categorized into five groups based on their intensity. 1: Peaks with intensities within the 0–20% maximum intensity range, 2: Peaks within the 20–40% intensity range, 3: Peaks within the 40–60% intensity range, 4: Peaks within the 60–80% intensity range, and 5: Peaks within the 80–100% intensity range.

# B ADDITIONAL BACKGROUND

## B.1 MORE DETAILS ABOUT CRYSTAL SYMMETRY

The seven crystal systems are cubic(#1), hexagonal(#2), tetragonal(#3), orthorhombic(#4), trigonal(#5), monoclinic(#6), and triclinic(#7). Crystals can be diagrammatically represented by an orderly stacking of lattice cells, whose shape determines the crystal system to which they belong. Unit cells of identical shape can have lattice points, representing an atom or group of atoms, at their centers or faces, in addition to the corners. These additional lattice points further divide the seven crystal systems into 14 Bravais lattices, which are then subdivided into 32 crystal classes, or point groups. Each point group corresponds to a possible combination of rotations, reflections, inversions, and improper rotations. When translational elements are included, these point groups yield the 230 space groups (Clegg, 2023).

## B.2 IN-LIBRARY AND OUT-OF-LIBRARY TASKS

In-library identification is a fundamental task in crystallography that aims to accurately identify crystal types based on XRD patterns measured in various environments. Since 1938, crystallographers have been documenting all discovered structures and archiving them as Powder Diffraction Files (PDFs). By comparing and retrieving these PDFs, researchers can determine the structures of studied materials by matching them with historical data. With advancements in computing, numerous software programs have been developed to assist in the search-match process.

A related concept is out-of-library identification, which involves discovering previously unknown structures that lack recorded data. This process depends on symmetry identification to determine the basic space group information, followed by subsequent refinement. Traditionally, the geometric characteristics of a crystal system are determined based on physical properties, such as electrical conductivity, optical behavior, and thermal properties. The space group is then identified by examining extinction effects. Next, the ideal chemical formula is determined based on the estimated number of atoms and Wyckoff positions. Validation is performed using techniques such as Rietveld refinement and site optimization.

Therefore, we provided two types of splits:

- **In-library Classification:** In our in-library benchmark, the dataset is divided based on different simulation environments. Both the training and testing datasets contain the same structures, but under varying simulation conditions. This setup corresponds to in-library identification in XRD phase analysis.

- **Out-of-library Classification:** In our out-of-library benchmark, the dataset is split according to crystal types. The training and testing datasets consist of different structures, mirroring the out-of-library identification process in XRD phase analysis.

## B.3 SEARCH-MATCH APPROACH

Conventionally, the primary method for in-library phase identification in X-ray powder diffraction patterns is the search-match approach, which consists of three main steps. Initially, (d, I) values, representing interplanar spacing (d) and intensity (I) of each peak, are extracted from the pattern. Subsequently, potential phases are sought from diffraction databases based on characteristic d values, such as Hanawalt indexes (University College London, date accessed). Following this, candidate phases are compared to the (d, I) values of the pattern using a scoring system to assess alignment, aiding in selecting the most suitable candidate phase. This process iterates until satisfactory alignment is achieved for most (d, I) values. A recent study (Lutterotti et al., 2019) employs an approach utilizing the Rietveld method for conducting a Full Profile Search-Match (FPSM). Crystal structures selected from databases are automatically aligned with raw data via the Rietveld method, culminating in a score determined by $R_{wp}$ (Millane, 1989). The phase with the highest score is acknowledged, and this iterative procedure persists until specified criteria are met. Subsequently, a Rietveld quantification is executed to ascertain the weight fraction of each phase. Despite advancements in computer technology that have enabled new qualitative methods, the core workflow of the search-match ap-

proach remains unchanged: matching experimental data with database entries, which contain partial information abstracted from raw diffraction patterns, and calculating a corresponding score.

## B.4  XRD PATTERN SIMULATION

Powder XRD provides a one-dimensional representation of three-dimensional diffraction patterns and is the most common experimental measurement, as shown in Figure 8. These patterns reflect the relative arrangement of atoms in three-dimensional space. The interaction of X-rays with condensed materials during diffraction involves elastic collisions between photons and electrons. Consequently, factors that affect atomic arrangement and electron behavior influence the shape of the diffraction pattern, emphasizing the need for high-fidelity simulation technologies. To address this gap, we developed Pysimxrd.

The diffraction vector $\mathbf{G}^*$ is defined as

$$|\mathbf{G}^*| = \frac{2\sin\theta}{\lambda},\tag{1}$$

where $\lambda$ is the wavelength of the scattered particle and $\theta = 2\theta/2$, with $2\theta$ being the angle between the incident and diffracted beams. In three dimensions, scattering occurs only at a discrete set of reciprocal vectors, $\mathbf{K}$, forming the reciprocal lattice,

$$\mathbf{K} = h\mathbf{a}^* + k\mathbf{b}^* + l\mathbf{c}^*,\tag{2}$$

where $\mathbf{a}^*$, $\mathbf{b}^*$, and $\mathbf{c}^*$ are the reciprocal lattice vectors, and $h$, $k$, and $l$ are constants. The diffraction condition is defined as

$$2\pi\mathbf{G}^* = \mathbf{K}.\tag{3}$$

The x-axis of simulation patterns $d$ is the reciprocal of $|\mathbf{G}^*|$, i.e., $d = 1/|\mathbf{G}^*|$.

The diffraction intensity $I$ on each diffraction vector $\mathbf{G}^*$ for a single phase is determined by

$$I(\mathbf{G}^*) = SF^*F\varnothing LPOD + I^{\text{BG}},\tag{4}$$

where $S$ denotes the scale factor, $F$ is the structure factor, $F^*$ is the complex conjugate of $F$, $\varnothing$ is the profile function, $L$ is the Lorentz-polarization factor, $P$ is the multiplicity, $O$ is the preferred orientation factor, $D$ is the Debye-Waller factor, and $I^{\text{BG}}$ is the background intensity. The structure factor is computed as:

$$F = \sum_{j=1}^{N} f_j e^{2\pi i\mathbf{G}^* \cdot \mathbf{R}},\tag{5}$$

where $f_j$ is the form factor Hubbell et al. (1974) in XRD, $\mathbf{R}$ is the lattice coordinate of atom $j$, and $N$ is the total number of atoms in a lattice cell.

The Lorentz-polarization $L$ is calculated by,

$$L = \frac{1 + \cos^2 2\theta}{\sin^2 \theta \cos \theta},\tag{6}$$

where $\theta = \arcsin\left(\frac{\lambda|\mathbf{G}^*|}{2}\right)$. The multiplicity $P$ is determined by counting the number of diffraction vectors present within an Ewald diffraction sphere. The Debye-Waller factor $D$ is calculated by,

$$D = e^{-2M},\tag{7}$$

where $M = \frac{6h^2 T}{mk\Theta^2}\left(\phi\left(\frac{\Theta}{T}\right) + \frac{\Theta}{4T}\right)\left(\sin^2\theta\right)/\lambda$, $h$ is Planck's constant, $m$ is atom mass, $k$ is Boltzmann constant, $\Theta$ is the average characteristic temperature, $T$ is absolute temperature, and $\phi(\Theta/T)$ is the Debye function.

The profile function ($\varnothing$) is modeled by convolving various factors, including diffraction, detector geometry, and noise factors etc. The simulated peaks are calculated by:

$$y(x) = W * G * S.\tag{8}$$

Here, * denotes the convolution process, and $W$, $G$, and $S$ represent the contributions to the observed XRD pattern from diffraction emission, instrumental factors, and the noise mixture, respectively. $S$ is modeled as a Gaussian peak.

The $W$ is a Voigt function Armstrong (1967),

$$W = \frac{1}{\sigma\sqrt{2\pi}} \int_{-\infty}^{\infty} \left[ \frac{\gamma}{(2\theta - t)^2 + \gamma^2} \right] \exp\left( -\frac{(2\theta - t)^2}{2\sigma^2} \right) dt. \tag{9}$$

Peak broadening is correlated with the Full Width at Half Maximum (FWHM, $\Gamma$), where $2\gamma = 2\sqrt{2\ln 2}\sigma = \Gamma$ Caglioti et al. (1958). $\Gamma$ is calculated by Scherrer's equation Miranda & Sasaki (2018) related to finite grain size.

The geometrical factorVan Laar & Yelon (1984), $G$, accounts for the actual dimensions of the detectors and the powder sample. It is defined as follows:

$$D(2\alpha, 2\theta) = \frac{\delta I(2\theta)}{\delta(2\alpha)} \, d(2\alpha), \tag{10}$$

$$G(2\alpha, 2\theta) = \frac{L}{4HSh\cos(2\alpha)} \int dz, \tag{11}$$

where $2\alpha$ represents the Bragg angle. The detector is slit-shaped, with a height of $2H$, and the sample has a height of $2S$. $L$ denotes the distance between the sample and the detector.

The simulation environments in our study encompass three key aspects. The first aspect pertains to the sample, where we simulate the average grain size, the ideal orientation effect of the powder, and the lattice cell's extinction and torsional deformation under internal stress. The second aspect involves the testing conditions, accounting for factors such as atomic thermal vibrations at room temperature, air scattering-induced signal noise, and instrumental vibration-induced noise. The third aspect relates to the diffractometer, including the zero shift of the angular position and detector's geometry.

The adjustable parameters of the simulation environments include:

- Finite grain size (for Voigt peaks): the average grain size is set as a random number in the range of 2 nm to 50 nm to replicate the degree of experimental broadening (Maniammal et al., 2017). Instrumental broadening becomes apparent in the experimental patterns as convolution with the crystal peak occurs. Typically, in the refinement process, it is common to separate observed peaks into Voigt and instrumental components. However, quantifying the instrumental component proves challenging as it varies with the diffraction angle. Therefore, in our simulations, we opt for a smaller grain size than realistic to accurately replicate the degree of experimental broadening. Pysimxrd can also model the asymmetric peak by convolving the instrumental geometry factors in equation 11. If a convolution peak is applied, the grain size range must be adjusted accordingly.

- Orientation: the ideal powder sample is assumed to have no orientation, however, this is impossible. The orientation effect arises from the uneven distribution of small grains in the incident beam, leading to an uneven distribution of reciprocal sites across the reciprocal spheres of the powder sample. To simulate the impact of orientation, we set the intensity by perturbation within a 40% range based on the ideal simulation to mimic the orientation randomness.

- Thermal vibration: the temperature is converted to the kinetic energy of atoms, which is simulated by allowing atoms to shift from their average positions within a range of 0.01-0.5 angstrom.

- Internal stress: internal stress is simulated by applying elastic deformation of the recorded lattice constant by up to 10%.

- Instrument zero shifting: the zero shift is simulated by randomly translating $2\theta$ within -1.2 to 1.2 degrees.

- Instrument noise: 2% Gaussian white noise is added to the entire pattern to reflect instrument noise.

- Inelastic Scattering: A sixth-order polynomial function is employed to simulate the background distribution, encompassing phenomena such as Compton scattering, fluorescence, and multiple scattering. This simulated background is then integrated into the XRD pattern with a 2% ratio.

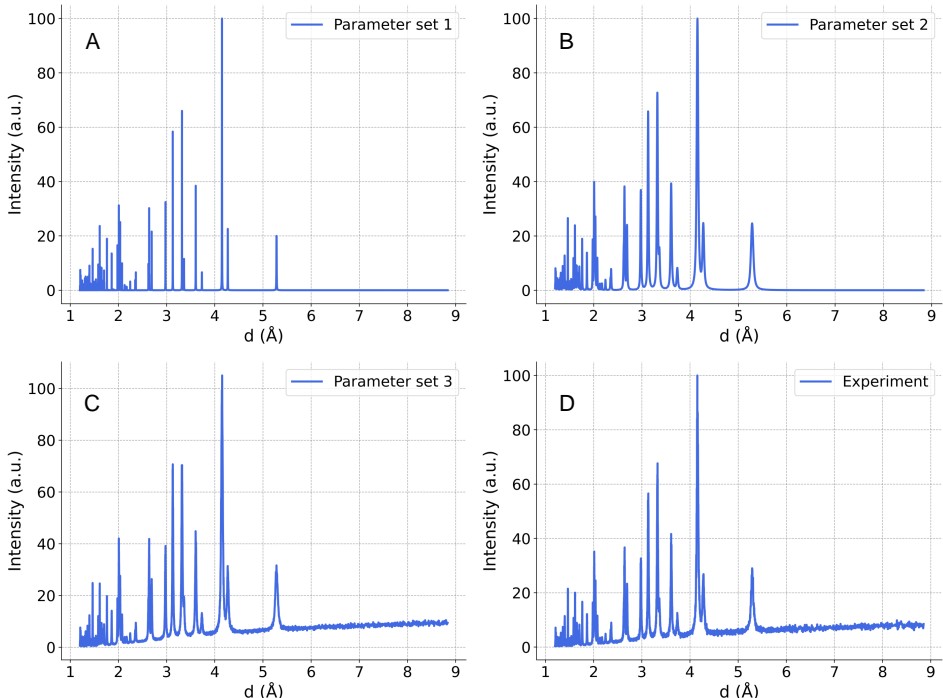

Figure 7: Comparison of simulation pattern with experimental pattern of $PbSO_4$ (A) The generated $PbSO_4$ powder XRD without considering the simulation environment. (B) The generated $PbSO_4$ powder XRD based on a set of parameters that account for finite grain size, orientation, thermal vibrations, and internal stress. (C) The generated $PbSO_4$ powder XRD with an extended set of parameters that further include zero-shift correction, noise, peak convolution, and background effects. (D) The experimentally observed data. Note: The $PbSO_4$ structure corresponds to entry 1010950.cif from the Crystallography Open Database. The parameter sets used in the simulations were optimized by minimizing the differences between the simulated and experimental patterns, rather than being randomly assigned. All patterns were converted to Q-space using a wavelength of 1.54 Å.

As illustrated in Figure 7, the multiphysical coupling observed in experimental XRD can be incorporated into the simulation parameter space by introducing more realistic conditions. Specifically, considering factors such as orientation, thermal vibrations, zero-shift corrections, noise, and other sources of randomness significantly improves the realism of the simulations. These parameters collectively influence the arrangement of peaks, their intensities, and overall shapes in the diffraction pattern.

### B.5 CRYSTAL AND DIFFRACTION

Among various investigative methods, diffraction analysis stands out as exceptionally potent for probing microstructure, primarily due to its sensitivity to atomic arrangement and the element specificity of atom scattering power. Each intensity maximum, termed a line profile or peak, within a diffraction pattern reflects the atomic arrangement along certain direction of the diffracting material.

Powder X-ray diffraction, in particular, furnishes a remarkably diverse array of structural details, encoded in the material- and instrument-specific distribution of coherently scattered monochromatic wave intensity, with wavelengths corresponding to lattice spacing. The X-rays are generated by a cathode ray tube, filtered to produce monochromatic radiation, collimated for concentration, and directed towards the sample. When conditions satisfy Bragg's Law, which relates the wavelength

of electromagnetic radiation to the diffraction angle and lattice spacing in a crystalline sample, the incident rays interact with the sample to produce constructive interference and a diffracted ray, as shown in Figure 8.

The diffracted X-rays are subsequently detected, processed, and quantified. By scanning the sample across a range of diffraction angles, all possible diffraction directions of the lattice are accessible due to the random orientation of powder sample.

## B.6   SYMMETRY RECOGNITION IN STRUCTURE DETERMINATION

Pattern refinement (Rietveld, 1967; 1969) is essential for accurately determining the crystal structure in diffraction studies. In XRD refinement, the initial step involves identifying the crystal phase of the given XRD pattern. This is crucial because it facilitates the calculation of the static structure factor (Svensson et al., 1980), which aids in determining atomic sites during refinement.

Using existing crystal databases, the commonly employed search-match approach retrieves potential structures that match the observed pattern, effectively performing in-library identification. The matched structure, along with basic crystallographic information such as crystal symmetry, serves as a foundational element for further analysis. In out-of-library situations, the geometry of the crystal lattice must be established from scratch, as there is no recorded structure history. By solving the diffraction indices corresponding to a set of lattice planes for each diffraction peak, a process known as indexing calculation (Rodriguez-Garcia et al., 2021), the crystal symmetry can be determined.

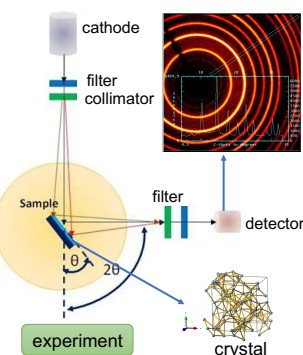

Figure 8: Experimental procedure for recording XRD patterns.

Whether the observed patterns are from in-library or out-of-library crystals, symmetry identification combined with indexing is a universal method for determining lattice geometry to initiate the refinement process. This is why many researchers seek a general method for deriving the symmetry of structures based on diffraction patterns. Once the fundamental crystallographic information is obtained, techniques such as pattern decomposition and Rietveld refinement (Rietveld, 1967; 1969; Le Bail, 2005) are employed. These methodologies help determine atomic sites, temperature factors, grain sizes, residual stress, and other intricate structural details from the diffraction patterns, thereby contributing to the comprehensive understanding of a crystal structure.

## B.7   WHOLE POWDER PATTERN FITTING AND REFINEMENT

Structure refinement is an effective method for determining crystal structures. Traditionally, refinement begins with a set of reasonably estimated parameters with physical significance. Accurately identifying the structure's space group aids in establishing rational parameters. However, without prior structural knowledge, this process becomes trial and error. Although identifying the space group initiates the process, its validation relies on refinement outcomes.

A significant advancement in powder diffraction fitting for structure refinement emerged with the Rietveld method(Rietveld, 1967; 1969), which pioneered whole pattern fitting over the analysis of individual, non-overlapping Bragg diffraction peaks. The Whole Powder Pattern Fitting (WPPF) approach within the Rietveld method uses profile intensity calculations and a least-squares algorithm for structure refinement. By minimizing the disparities between observed experimental profiles and theoretical profiles, it extracts all structural information. Once fitting reaches its limit, the parameters in the theoretical model represent the only determinable physical information. SimXRD's pattern simulation also relies on the structural information contained within the theoretical model. The Pawley(Pawley, 1981) and Le Bail(Le Bail et al., 1988) methods, two widely used WPPF techniques, are developed after the Rietveld method.

These parameters with physical significance are diverse, and one important type is peak shape parameters. The parameters of each diffraction peak (positions, intensities, and shapes) within a whole XRD pattern offer detailed structural insights from different aspects, making their quantitative and

accurate extraction from the overall XRD profile highly significant. This quantification of peak information and the subsequent derivation of physical parameters constitute the essence of the refinement endeavor.

## B.8    SIM2REAL GAP

To illustrate the advantages of Pysimxrd in generating high-fidelity powder XRD patterns, we compared its performance with a widely recognized software, GSAS-II (Toby & Von Dreele, 2013). The simulation process involves the use of refinement software, starting with the determination of suitable parameter sets through the refinement procedure provided by each software. The lattice constants derived from these refinements are summarized in Table 7. Once the parameters are confirmed, the corresponding simulation patterns are generated.

Table 7: Derived crystal structures in refinement.

| Models | Crystals | Lattice Constants (Å) | | | | | |
|---|---|---|---|---|---|---|---|
| | | a | b | c | $\alpha$ | $\beta$ | $\gamma$ |
| Pysimxrd | $Mn_2O_3$ | 9.4082(2) | 9.4082(2) | 9.4082(2) | 90 | 90 | 90 |
| Pysimxrd | $RuO_2$ | 4.5381(7) | 4.5381(7) | 3.1248(4) | 90 | 90 | 90 |
| GSAS-II | $Mn_2O_3$ | 9.4197(1) | 9.4197(1) | 9.4197(1) | 90 | 90 | 90 |
| GSAS-II | $RuO_2$ | 4.5351(8) | 4.5351(8) | 3.1226(0) | 90 | 90 | 90 |

Two experimental powder XRD patterns were obtained using an X'Pert Pro MPD (Panalytical) in Bragg-Brentano geometry with a copper X-ray source ($\lambda_{\text{K-}\alpha} = 1.5406$ Å). The samples used were $Mn_2O_3$ and $RuO_2$ crystals. As shown in Figure 9, Pysimxrd and GSAS-II demonstrated comparable accuracy in reconstructing the $Mn_2O_3$ pattern. However, Pysimxrd showed significant advantages in reconstructing the $RuO_2$ pattern. This superiority stems from Pysimxrd's ability to account for a broader range of simulation environments and its greater flexibility in tuning parameters, which resulted in a more accurate reconstruction with lower R-factors.

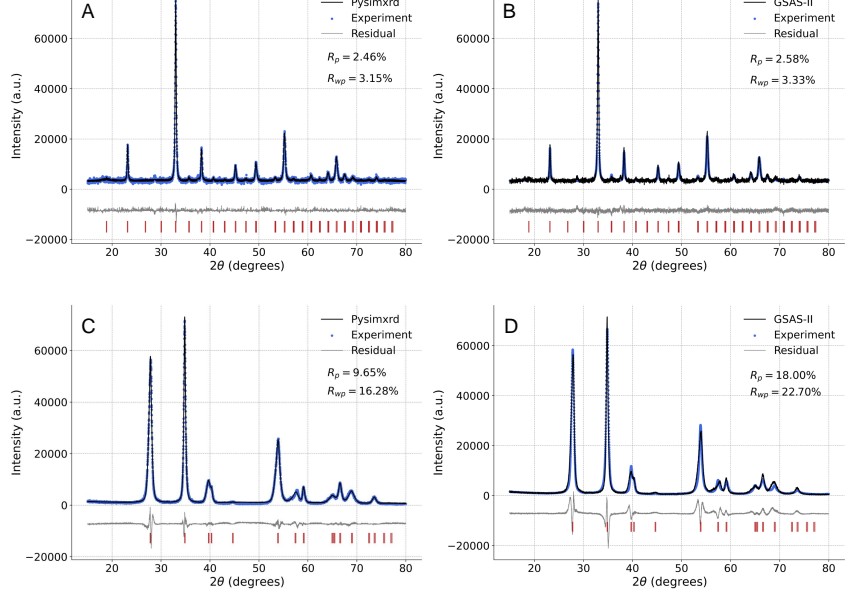

Figure 9:    Comparison of simulated and experimental patterns: (A) The $Mn_2O_3$ pattern simulated using Pysimxrd. (B) The $Mn_2O_3$ pattern simulated using GSAS-II. (C) The $RuO_2$ pattern simulated using Pysimxrd. (D) The $RuO_2$ pattern simulated using GSAS-II. The R-factors, calculated as (Toby, 2006), are used to evaluate the quality of the fitting.

## B.9 THE T-SNE PLOTS OF POWDER XRD PATTERNS

We also computed t-SNE plots (Van der Maaten & Hinton, 2008) based on XRD patterns, along with their corresponding crystal systems and space groups, as shown in Figure 10. The results indicate that the projection plots for both SimXRD and RRUFF exhibit bilateral symmetry and distinct manifolds. However, due to the limited data in RRUFF, the two datasets show some discrepancies, reflected in the differences in crystal system and space group distributions across the XRD patterns.

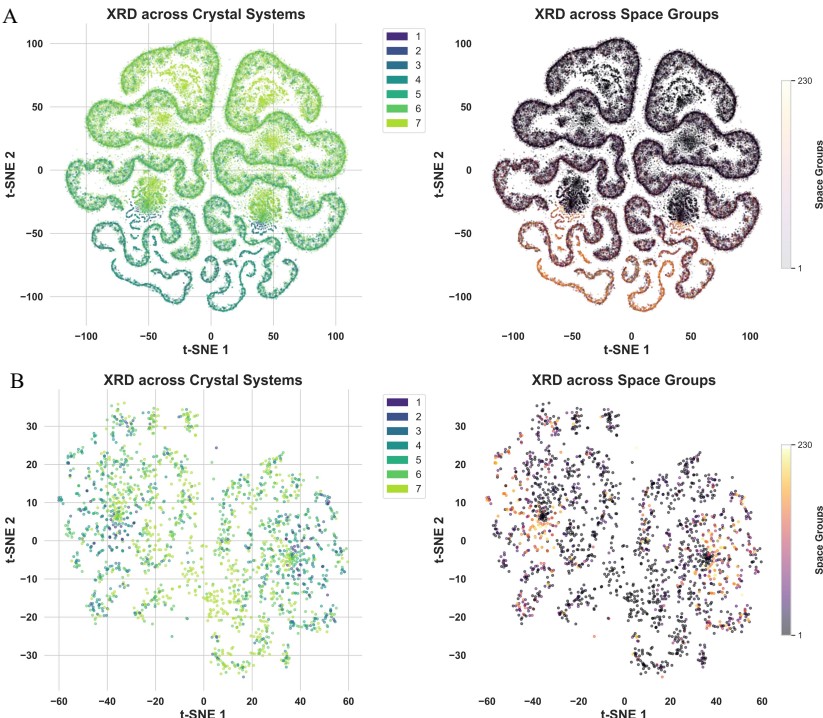

Figure 10: (A)The t-SNE plot for XRD patterns in the SimXRD validation dataset. (B) The t-SNE plot for XRD patterns in the RRUFF dataset. Numbers 1–7 represent the seven crystal systems (Appendix B.1), while 1–230 correspond to the space groups.

