# OpenReview forum: "SimXRD-4M: Big Simulated X-ray Diffraction Data and Crystal Symmetry Classification Benchmark"
_ICLR.cc/2025/Conference — ICLR 2025 Poster_

### Official Review · Reviewer_UoDH · 2024-10-21

**Soundness:** 3
**Presentation:** 2
**Contribution:** 3
**Rating:** 8
**Confidence:** 4

**Summary:**

Overall, the work presents a valuable dataset and simulation environment for PXRD patterns, which is a great topic of interest in the materials science community.

**Strengths:**

- Accurate simulation of PXRD patterns is a timely topic of great recent research interest, and actually something I have grappled with in my own work, so I was pleased to read of your work that systematically does this in an easy software package.
- Allows users to simulate PXRD patterns with a wide variety of realistic settings not offered in other packages: grain size, stress, temperature, orientation, background, drift, noise. This is very important for addressing the sim2real gap in computational materials science.
- Some really cool figures (Figure 1, Figure 2) that show that we can simulate PXRD patterns that look like real experimental data. Other leading packages like Pymatgen are known to be unable to do this.
- Validates that models trained on these simulated PXRD patterns can generalize to experiments on real data.
- Well-presented and easily usable open-source code, especially this early in the review process. It gives me faith that this is a reproducible, adaptable contribution to the community.
- Brings to light the weaknesses of previous works (CNN-based) in dealing with the unbalanced symmetry classification task. This is especially important, as it would presumably be more important to analyze the less common symmetry groups.

**Weaknesses:**

I have a number of important suggestions for improvement, but most of these are easily addressable, rather than fundamental limitations of your work:
- Baseline for Dataset: This work is under the datasets and benchmarks task. You certainly addressed the benchmark part of this, by evaluating multiple space group classification models on your work. However, it is necessary to have baselines for the dataset. In particular, if you simulated PXRD patterns with Pymatgen and GSAS-II [6] and trained models on these, how would it perform on experimental data, as compared to training on SimXRD? The value of the dataset presumably comes from the fact that training on it would reflect real-world conditions with better fidelity, and thus, lead to boosted performance in testing, as compared to training on less realistic patterns. Thus, you need to show that other available datasets/simulators (Pymatgen, GSAS-II) cannot lead to the same sim2real transfer.
- Presentation of Figures: It's really cool that you can simulate PXRD patterns that look like actual experimental data. I'd like to see some more examples of these in a larger diagram. I only saw some small pictures in Figure 1 and Figure 2. I think displaying 4-6 examples of these in a large figure (preferably in the main text) would really drive home the point that your work provides the most realistic simulations of PXRD patterns available. In line with the previous comment, you should also compare to what other simulators (Pymatgen, GSAS-II) can give.
- Simulation Fidelity Evaluation: Can you take maybe 10 experimental patterns with known ground truth structures, and simulate patterns with SimXRD, Pymatgen, and GSAS-II? Then see the R_w goodness of fit value as compared to the experimentally observed? Of course, in this experiment, you'd need to play with the parameters (if available) to get the best fit possible from each software (although I think only GSAS-II, SimXRD, and possibly TOPAS, if you have that; make this possible). I see that you already have the embryo of this in Figure 1, so hopefully it should not be that hard to do on a few patterns. Again, this would emphasize the point that you have the most realistic simulator available.
- Address Structure Solution: Although symmetry classification is important, it's just one step towards the grand challenge of end-to-end structure determination from PXRD. This end-to-end solution paradigm has gained a lot of interest, as of 2024. You seem not to cite the relevant works in the field: please look at them and cite them [1, 2, 3, 4, 5]. It's probably not feasible to train or evaluate such generative structure prediction models in such a short period of time, especially since most of them are very new and not-yet open source. But at least mention them, because I think in the future, this dataset will be very valuable for training the PXRD structure solvers, and you should let the relevant authors see that your work is relevant to them, so that the impact of your work can be maximized!
- Writing: It seems that your intro is fixated on the benchmark part of your contribution, but I think you can emphasize more how this dataset itself is valuable. Again, in my own experience, I have actually been looking for a dataset like this, to address the sim2real gap common in computational crystallography. For instance, out of the bullets on the bold bulleted list at the end of page 2, I actually find the third ("Experimental Data Generalization") more interesting, and I think you could emphasize that direction more in your presentation.

[1] Guo, G., Goldfeder, J., Lan, L., Ray, A., Yang, A. H., Chen, B., ... & Lipson, H. (2024). Towards end-to-end structure determination from x-ray diffraction data using deep learning. npj Computational Materials, 10(1), 209.

[2] Lai, Q., Yao, L., Gao, Z., Liu, S., Wang, H., Lu, S., ... & Ke, G. (2024). End-to-End Crystal Structure Prediction from Powder X-Ray Diffraction. arXiv preprint arXiv:2401.03862.

[3] Guo, G., Saidi, T., Terban, M., Billinge, S. J., & Lipson, H. (2024). Diffusion Models Are Promising for Ab Initio Structure Solutions from Nanocrystalline Powder Diffraction Data. arXiv preprint arXiv:2406.10796.

[4] Li, Q., Jiao, R., Wu, L., Zhu, T., Huang, W., Jin, S., ... & Chen, X. (2024). Powder Diffraction Crystal Structure Determination Using Generative Models. arXiv preprint arXiv:2409.04727.

[5] Riesel, E. A., Mackey, T., Nilforoshan, H., Xu, M., Badding, C. K., Altman, A. B., ... & Freedman, D. E. (2024). Crystal structure determination from powder diffraction patterns with generative machine learning. Journal of the American Chemical Society.

[6] Toby, B. H., & Von Dreele, R. B. (2013). “GSAS-II: the genesis of a modern open-source all purpose crystallography software package”. Journal of Applied Crystallography, 46(2), 544-549. doi:10.1107/S0021889813003531

**Questions:**

- Are you able to generate PXRD patterns of varying Q-ranges and/or x-ray wavelengths?

---

> ### Author Response · Authors · 2024-11-20
>
> Thank you for recognizing our work. We have carefully addressed your comments and made corresponding improvements to our manuscript. As you acknowledge the significance of our work, we kindly request your endorsement and consideration of a higher score, should our revisions meet your satisfaction.
>
> >Weaknesses-Baseline for Dataset: In particular, if you simulated PXRD patterns with Pymatgen and GSAS-II [6] and trained models on these, how would it perform on experimental data, as compared to training on SimXRD?
>
> Thank you for your comment. As you suggested, the generative ability for testing on experimental data is crucial. However, due to time constraints, regenerating a large database based on MP using tools like Pymatgen, GSAS-II, and recalculating benchmarks is highly challenging. We have added further clarification in **lines 153-160** to emphasize this point, as summarized below:
>
> "Earlier studies primarily relied on widely-used software tools, such as
> Pymatgen (Ong et al., 2013), FullProf (Rodr´ıguez-Carvajal, 2001), and GSAS-II (Toby & Von Dreele, 2013), to simulate patterns for training. However, these approaches often showed varying degrees of performance drop when applied to experimental test sets. SimXRD addresses these limitations by employing the newly developed PysimXRD, which accounts for a wider range of real-world conditions."
>
> >Weaknesses-Presentation of Figures: Displaying 4-6 examples of these in a large figure (preferably in the main text).
>
> Thank you for your suggestion. Figure 2 demonstrates that, with a carefully selected set of parameters (optimized according to the experimental pattern), the simulated pattern aligns well with the experimental one.
>
> We taken your have advice and added a new section in Appendix B.9 SIM2REAL GAP (**lines 1181-1232**), which provides additional cases for PysimXRD and GSAS-II. The parameter searching process requires a refinement function, so we have only compared our approach with GSAS-II. (Due to the page limitations of the ICLR conference, we did not present this in the main text.)
>
> >Weaknesses-Simulation Fidelity Evaluation: Can you take maybe 10 experimental patterns with known ground truth structures, and simulate patterns with SimXRD, Pymatgen, and GSAS-II? Then see the R_w goodness of fit value as compared to the experimentally observed.
>
> Thank you for your comment. We have taken your advice and added an additional section in Appendix B.9 SIM2REAL GAP (**lines 1181-1232**), providing a comparison between PysimXRD and GSAS-II. We present the experimental patterns, refined crystal constants, and fitting Profile factors.
>
> >Weaknesses-Address Structure Solution: You seem not to cite the relevant works in the field: please look at them and cite them [1, 2, 3, 4, 5].
>
> Thank you for your suggestions. We have added the latest research to our paper, as shown below:
>
> "More recent studies (Guo et al., 2024a;b; Li et al., 2024; Riesel et al., 2024; Lai et al., 2024) on powder XRD for crystal prediction and generation have employed advanced ML architectures, achieving significant progress and noteworthy results."
>
> >Weaknesses-Writing: I think you can emphasize more how this dataset itself is valuable.
>
> Thank you for the comment.
>
> * We have emphasized the challenge of sim2real in the revised version as follows: (**lines 073–079**) "The limited scale of experimental data presents two significant limitations in this field : (1) insufficient data to comprehensively test generative capabilities and (2) the difficulty of effectively utilizing small-scale experimental datasets for model tuning and transfer learning.
>
> * We have also enhanced the "Experimental Data Generalization" section (**lines 153-160**), summarized as follows: "Earlier studies primarily relied on widely-used software tools, such as Pymatgen (Ong et al., 2013), FullProf (Rodr´ıguez-Carvajal, 2001), and GSAS-II (Toby & Von Dreele, 2013), to simulate patterns for training. However, these approaches often showed varying degrees of performance drop when applied to experimental test sets. SimXRD addresses these limitations by employing the newly developed PysimXRD, which accounts for a wider range of real-world conditions."
>
> >Questions: Are you able to generate PXRD patterns of varying Q-ranges and/or x-ray wavelengths?
>
> Yes, we can. SimXRD is simulated within specific Q-ranges, as it is designed to function as a standard database. In contrast, PysimXRD is open-source, allowing for high-fidelity simulations across arbitrary diffraction angles or Q-ranges.

---

> > ### Author Response · Authors · 2024-11-25
> > **A kind remind of author-reviewer discussions**
> >
> > Thanks for your contributions to the reviewing process. As the deadline for the author-reviewer discussion approaches, we kindly request your feedback on whether our responses have satisfactorily addressed your concerns. Should you have any additional suggestions or comments, please do not hesitate to share them with us. We would be more than willing to engage in further discussions and make any necessary improvements.
> >
> > Thank you once again for dedicating your valuable time to reviewing our work.

---

> ### Comment · Reviewer_UoDH · 2024-11-26
> **Thanks**
>
> I appreciate your revisions, especially Figures 7 and 9, that show that you can achieve realistic PXRD patterns that adhere to experimental observations. As such, I will raise the score. Looking forward to seeing this work in print.

---

> > ### Author Response · Authors · 2024-11-27
> > **Appreciate your recognition**
> >
> > We sincerely appreciate your recognition, which greatly encourages our research efforts.
> > Thank you for the constructive suggestions that have significantly improved the quality of our paper.
> > Your essential contribution to enhancing our work is deeply valued.

---

### Official Review · Reviewer_xPRn · 2024-10-22

**Soundness:** 3
**Presentation:** 3
**Contribution:** 3
**Rating:** 5
**Confidence:** 4

**Summary:**

This paper presents SimXRD, a simulated X-ray diffraction (XRD) pattern dataset for benchmarking crystalline symmetry classification tasks. The patterns are simulated using PySimXRD, a custom software based on diffraction physics, for (filtered) crystal structures in the Materials Project database. Using the dataset, analyses of crystal symmetry data and comparisons of ML models are performed.

**Strengths:**

The presented large-scale simulated XRD dataset is open-sourced, which could benefit the community.

**Weaknesses:**

- The main contribution of this work is unclear. The analyses and benchmarks do not provide much insight into ML research. If the PySimXRD software contains new/better functions than previous XRD simulators, they should be specified; even so, the contributions are in physics or materials science, not ML. Rather than ICLR, venues like *Scientific Data* seem more suitable for this paper.
- The presentation logic and clarity of this paper are poor. Examples are listed in Questions.

**Questions:**

- The simulated dataset uses lattice plane distance as the x-axis. How is it related to 2$\theta$ which is more commonly used in XRD data, and what’s the reason for this choice?
- The generalizability of models to experimental data is assessed by training on SimXRD and testing on RRUFF. Is RRUFF representative of experimental data influenced by various factors (materials type, physical condition, equipment, etc.)? How do the authors ensure this is a rigorous assessment?
- Line 157, “almost all related research…due to the lack of easily accessible datasets”, what does this mean?
- Line 345, is “being biased by classes of high frequency” a problem of CNN? It should be more related to the objective/loss function.
- Regarding presentation clarity:
  - Line 48, structure analysis being challenging for domain experts is irrelevant to the search-match method.
  - Line 59, what challenges do introducing large-scale databases present?
  - Line 79, repeated words.
  - Line 216, the details on the Materials Project are irrelevant to this work.

---

> ### Author Response · Authors · 2024-11-20
>
> Thank you for reviewing our paper and providing feedback. We have responded to all of your comments comprehensively and welcome further discussion to clarify and resolve these matters.
>
>
> >Weaknesses: The main contribution of this work is unclear. The analyses and benchmarks do not provide much insight into ML research.
>
> Thank you for your comment. We would like to emphasize that our insight is highly related to ML research:
> * We provide comprehensive, high-fidelity datasets for fundamental data-driven model developments.
> * We benchmarked 21 different ML models, including those developed specifically for PXRD data and others for sequence data, offering a broad and systematic evaluation.
> * We highlight the long-tailed distribution in symmetry classification, which is an ML problem. We evaluate the model performance under different objective functions to show that addressing it improves the model performance.
> * We study the model generalization ability to different experimental factors and crystals. Identifying such problems is vital for the ML community.
>
> As recognized by other reviewers, "enabling advancement in crystallographic machine learning research; can be a useful resource for the community."（`Z7kM`）, "a strong motivation and need for this paper in the research community" (`Qfc3`), "a reproducible, adaptable contribution to the community (`UoDH`), the contribution of our work is clear and highly relevant to the ML community.
>
> >Questions 1: The simulated dataset uses lattice plane distance as the x-axis. How is it related to 2 theta, which is more commonly used in XRD data, and what’s the reason for this choice?
>
> Thank you for your comment. In spectroscopy, the d-spacing (d) representation is preferred over 2-theta. The d-I pattern is independent of the wavelength of the incident X-rays, making it universally applicable across all experimental X-ray source. In contrast, 2-theta patterns are comparable only when the incident X-rays share the same wavelength. This inherent limitation does not apply to d-I patterns, which offer greater versatility.
>
> If a 2-theta pattern is required, a d-I pattern can be easily converted using the wavelength of the incident X-rays and the fundamental Bragg equation.
>
> >Questions 2: The generalizability of models to experimental data is assessed by training on SimXRD and testing on RRUFF. Is RRUFF representative of experimental data influenced by various factors (materials type, physical condition, equipment, etc.)? How do the authors ensure this is a rigorous assessment?
>
> RRUFF is a representative experimental database for testing model performance, as it is one of the largest experimental databases currently available. To ensure a rigorous assessment, we train models on simulated XRD patterns without any observation of the experimental patterns, thus their performance on RRUFF reflects their generalization ability to corresponding experimental patterns.
>
> >Questions 3: Line 157, “almost all related research…due to the lack of easily accessible datasets”, what does this mean?
>
> Thanks for the question. What we mean is that the study of powder XRD currently lacks a broad community. Our work seeks to engage **diverse research communities by offering open-source data and ML workflows**, thereby fostering model innovation and advancing technological development in this area.
>
> >Questions 4: Line 345, is “being biased by classes of high frequency” a problem of CNN? It should be more related to the objective/loss function.
>
> Thanks for the question. Both model architectures and training strategies are crucial for modeling imbalanced class learning. As shown in Figure 5 of the revised pdf, although CNN11 is trained with the CNN models, it performs better on low-frequency crystals.
>
> >Questions 5: Line 48, structure analysis being challenging for domain experts is irrelevant to the search-match method.
>
> Thank you for your question. We emphasize that symmetry identification is the first and most critical step in structural analysis.
>
> >Questions 5: Line 59, what challenges do introducing large-scale databases present?
>
> Thanks for your question. Introducing large-scale databases presents challenges related to data quality, consistency, and accessibility. Our work provides a standardized ML workflow for powder XRD databases, with a comprehensive benchmark for model architecture comparison, model training, and model innovation to facilitate the development of this field.
>
> >Questions 5: Line 79, repeated words.
>
> Thanks for the comment. We modified the expression.
>
> >Questions 5: Line 216, the details on the Materials Project are irrelevant to this work.
>
> Thanks for the comment. Any XRD simulation database is based on crystal databases. The end-to-end inference process involves deriving crystal information from XRD patterns. The details of crystal databases provide a thorough understanding of the quality of the database and the task settings.

---

> ### Comment · Reviewer_xPRn · 2024-11-22
>
> Thanks for the response. I am still doubtful about the main contribution, but I acknowledge that the work is solid and comprehensive. Some of my concerns are not addressed:
> - Q2: how is the claim "RRUFF is a representative experimental database for testing model performance" supported? One possible way is to assess the data distribution of RRUFF. Otherwise, this statement is not rigorous (not that it's wrong, but it's an overclaim).
> - Q3: "almost all related research…due to the lack of easily accessible datasets" is logically strange. The research in this field is driven by a need in materials science to decode XRD data efficiently; dataset is a tool, not the goal.
>
> Nonetheless, I appreciate the time and effort the Authors put into improving the paper. I will raise my rating incrementally.

---

> > ### Author Response · Authors · 2024-11-23
> > **Response to suggestions**
> >
> > We sincerely thank you for actively participating in the discussion and helping us improve the quality of our paper. Based on your suggestions, we have made the following revisions:
> >
> > >Q2: Overclaiming RRUFF's representativeness
> >
> > We appreciate you pointing this out. In the revised version, we made four changes:
> >
> > * (Lines 75-76) We clarified that, due to the limitations of experimental patterns, it remains challenging to comprehensively test the generative capabilities of models.
> > * (line 107) We stated that the conclusions regarding experimental data are based on tests conducted using the RRUFF dataset.
> > * (Lines 473) In the benchmark results for generalization testing, we noted that the results were derived from the RRUFF dataset.
> > * (Lines 1290-1338) A new section (Appendix B.10) was added to compare the XRD distributions from SimXRD and RRUFF, providing insights into their differences and consistencies.
> >
> > >Q3: Expression problem
> >
> > Thank you for bringing this to our attention. In the latest version, we have removed the problematic sentence.
> >
> >
> > Once again, we greatly appreciate your valuable contribution to enhancing the quality of our paper.

---

> > ### Author Response · Authors · 2024-11-25
> >
> > We hope that our response addresses all of your concerns. If any problems remain to be addressed, please feel free to raise them; we will gladly answer them for you. If you have no further questions regarding our response, we would appreciate it if you could support our paper.
> >
> > Thank you once again for dedicating your valuable time to reviewing our work.

---

> > > ### Author Response · Authors · 2024-11-30
> > >
> > > Dear reviewer,
> > >
> > > Thanks for your contributions to the reviewing process. We kindly request your feedback on whether our responses have satisfactorily addressed your concerns. Should you have any additional suggestions or comments, please do not hesitate to share them with us. We would be more than willing to engage in further discussions and make any necessary improvements.
> > >
> > > Thank you once again for dedicating your valuable time to reviewing our work.

---

> > > > ### Comment · Reviewer_xPRn · 2024-12-02
> > > >
> > > > Dear authors,
> > > > Your revisions have addressed my remaining concerns well. I will raise my rating for Presentation.

---

> ### Author Response · Authors · 2024-12-02
> **Thank you for raising rating**
>
> Dear reviewer xPRn,
>
> We sincerely appreciate your reply and are glad to hear that we have addressed your remaining concerns well.
>
> Therefore we kindly ask you to consider lifting overall ratings.
>
> Thanks for your feedback again.
>
> Best regards,
>
> Authors.

---

> > ### Comment · Reviewer_xPRn · 2024-12-02
> >
> > As I have stated in my original review and previous response, I am still doubtful about the work's contribution and alignment with ICLR scope. I appreciate the improvement in technical soundness and presentation clarity, but I cannot lift the overall rating due to that.

---

> > > ### Author Response · Authors · 2024-12-04
> > >
> > > Dear Reviewer,
> > >
> > > Thanks for your reply. Although we have some differences in opinion of the ICLR scope, we sincerely thank you for your suggestions and comments during the discussion, which have helped us improve the quality of the paper. We greatly appreciate your recognition of our work.

---

### Official Review · Reviewer_Qfc3 · 2024-11-02

**Soundness:** 2
**Presentation:** 3
**Contribution:** 2
**Rating:** 6
**Confidence:** 4

**Summary:**

Introduces a new dataset of simulated X-ray diffraction dataset. It is generated with a novel technique based on known physical interactions. This dataset is the largest symmetric dataset to date. They train existing model architectures on their dataset, and show that some models have superior performance using their dataset. However, CNNs (one of the most commonly used architectures for this task) do not perform well. They also struggle to handle low-frequency class labels, even with re-weighting of sample importance.

**Strengths:**

The novel technique for generating samples of data is quite interesting; gathering experimental x-ray diffraction data on scale for large machine learning projects is difficult. This gives a strong motivation and need for this paper in the research community. The explanation for the techniques (in appendix B4) is very strong and extremely easy to read. Overall, the work and explanation provided in the appendix is one of the strongest parts of the paper, and is excellent in both content and writing quality.

The testing and the results appear to have had a significant amount of effort put into them and cover a variety of models. This provides strong support for the claims made and the generalizability of the results.

**Weaknesses:**

Overall: I think that this submission is on a very good track. As someone who has worked with x-ray diffraction data before, I understand the need for this work. However, the limitations of the dataset are too severe for practical use, and so I do not think that it is ready for use.

The two main weaknesses are:

A - There is no validation of SimXRD to show that the patterns it creates match reality. The results from the machine learning models trained on the data show that to ML algorithms it seems close enough, but analytical results would be nice. The only justification provided is the technique used to create the data and the results from training with SimXRD then testing on other datasets.

B - The poor performance of the models on the long-tailed distribution. These results undercut the quality of the distribution; they show that the dataset does not allow for training of models that perform accurately.

C - I do not understand Figure 2. There are just lots of images, with very little explanation. Since it is referenced as a primary part of the paper and does not have extensive description in the text, I do not understand what it is describing.

D - Overall writing clarity (outside of the appendix) could use some improvement. Explanation seems like it would be sufficient for someone already familiar with the work, but not somebody learning about this for the first time. Additionally, there are some grammatical mistakes which cause confusion (missing three commas in the sentence on line 69-70 for example).

**Questions:**

1 - In line 36: Transitional methods for what? I assume symmetry identification, though extra clarity would be appreciated.

2 - Line 156 claims that existing symmetry methods rely on 1D CNNs? As it is currently written, it is saying that every technique that has been introduced uses 1D CNNs, which contradicts lines 296-308.

3 - Lines 296-308: is this supposed to be a comprehensive list of techniques? As it is written, that is what it seems to be saying.

---

> ### Author Response · Authors · 2024-11-20
>
> We sincerely thank you for recognizing the significance of our work. We have carefully addressed your concerns one by one to improve the quality of this paper. If you are satisfied with the revisions, we kindly request your approval by considering an improved score.
>
> >Weaknesses A: There is no validation of SimXRD to show that the patterns it creates match reality.
>
> Thank you for your comment. We provide evidence of the fidelity of SimXRD in two aspects:
> 1. The experimental generative ability of SimXRD. Our results show that the performance of ML models trained on SimXRD and tested on RRUFF yields very consistent results.
> 2. In the revised version, we provide a more comprehensive analysis of the Sim2Real gap.
> * In Appendix B.4 XRD Pattern Simulation (**lines 1033–1070**), we present a detailed theoretical framework for data generation.
> * Figure 6 illustrates how the multiphysical coupling observed in experimental XRD can be incorporated into the simulation parameter space by introducing more realistic conditions.
> * In Appendix B.8 Sim2Real Gap (**lines 1182–1232**), we include case studies that compare the fidelity of pattern generation with GSAS-II. The results demonstrate that PysimXRD has the capability to generate high-fidelity simulated patterns.
>
> >Weaknesses B: The poor performance of the models on the long-tailed distribution.
>
> Thank you for your comment. Several baselines have shown promising results in our testing. The benchmark results indicate that CNN11 demonstrated strong performance in both in-library and out-of-library crystal system classification. Additionally, our findings suggest that PathTST and bidirectional time-series models (e.g., Bidirectional-GRU) show potential for studying XRD data, achieving over 90% accuracy for crystal system classification and 80% for space group classification. Furthermore, the results highlight that label smoothing and focal loss can address the long-tailed distribution of crystal systems to a certain extent, which may warrant further investigation.
>
> >Weaknesses C: I do not understand Figure 2.
>
> We apologize for the confusion. We have added a detailed caption to provide further clarification for Figure 2. Specifically, we add (**lines 235-243**):
>
> * Figure A: The transformation from a crystal structure to an XRD pattern involves considerations such as finite grain size, specific orientation, thermal vibrations, zero-shift corrections, and the scattering background. The recorded patterns are represented in Q-space, with lattice distances sorted along different diffraction directions. Using the crystal database, PysimXRD generates simulated XRD patterns for ML training, enabling intelligent symmetry identification tasks.
> * Figure B: We compared experimental measurement patterns with simulated XRD patterns generated using PysimXRD for a Li-rich layered oxide cathode structure, primarily composed of Li2MnO3. The simulation incorporates multiphysical coupling, resulting in generated patterns that align closely with experimental measurements, exhibiting small residual error.
>
> >Weaknesses D: Overall writing clarity (outside of the appendix) could use some improvement.
>
> Thank you for your suggestions. We have revised the manuscript to enhance its overall readability.
>
>
> >Questions 1: In line 36: Transitional methods for what? I assume symmetry identification, though extra clarity would be appreciated.
>
> Thanks for the suggestion, we have revised it as follows:
>
> "Traditional methods of symmetry identification involve a search-and-match process (Altomare et al., 2008)."
>
> >Questions 2: Line 156 claims that existing symmetry methods rely on 1D CNNs? As it is currently written, it is saying that every technique that has been introduced uses 1D CNNs, which contradicts lines 296-308.
>
> Thank you for your comment. Most models developed for PXRD are based on 1D CNNs. In lines 296-308 (Original version), other benchmarks discussed in this paper include various models developed for sequence data, rather than specifically for PXRD data. We have revised this sentence to be more rigorous:
>
> "Most existing methods **for symmetry identification** rely on one-dimensional convolutional neural networks to build classification models."
>
> >Questions 3: Lines 296-308: is this supposed to be a comprehensive list of techniques? As it is written, that is what it seems to be saying.
>
> Thanks for the question. It consists of all the existing symmetry classification models (i.e., CNN-based models). Moreover, we also evaluate advanced machine learning models, including recurrent models and transformers, that have not been adopted for such tasks. To the best of our knowledge, our benchmark considers a comprehensive list of techniques in the domain of symmetry classification.

---

> ### Comment · Reviewer_Qfc3 · 2024-11-21
> **Response to the First Rebuttal**
>
> Regarding updates to Figure 2 (A, B):
> Thank you for the update and caption. What confuses me still is the flow of “information” through the system. I see labeled images like: “crystal”, “reciprocal space”, etc, but I do not understand where the process starts, and how it precedes until completion. There are three red arrows, along with 3 green arrows, but they do not create a narrative. Something in the form: “image 1” -> “image 2” -> “image 3” -> … is the way I see similar diagrams made in literature. Since this diagram is one of the primary explanations for the contribution of the paper, further refinement and clarity of it is vital to the paper.
>
>
>
> ——— Satisfied Concerns ———
>
> Regarding Sim2Real gap:
> Thank you of the updates. This completely satisfies my concerns about this area.
>
> Regarding long-tailed distribution performance:
> This explanation satisfies my concerns about this area of the paper, so long as it is noted that this is an area requiring future exploration.
>
> Regarding the answers to the two questions:
> I am satisfied with the response to each, and I have no follow up questions.
>
> ——— Score Update ———
>
> I have updated the score incrementally for now. I have changed it from a 5 to a 6.

---

> > ### Author Response · Authors · 2024-11-22
> > **Thank you**
> >
> > We sincerely appreciate your valuable suggestions and recognition of our work's significance.
> >
> > In the updated version, we revised the figure according to your recommendations by decomposing it into three subfigures and providing detailed captions for each panel.
> >
> > Thank you once again for your support and feedback.

---

> > > ### Comment · Reviewer_Qfc3 · 2024-11-22
> > > **Response to Author Message Titled "Thank You"**
> > >
> > > I thank the authors for their work and updates to the figure. I believe it still needs further refinement, as it is still not entirely clear what is happening. Specific examples include:
> > > 1 - in part A, what is the starting point of this process? What is the ending point?
> > > 2 - in part C, what is the relationship between the information in the five panels?
> > >
> > > I believe my score of a 6 remains appropriate for the most recent version of the paper. I thank the authors for their work and wish them the best of luck in continuing research along this line.

---

> > > > ### Author Response · Authors · 2024-11-23
> > > > **Further refined Figure 2**
> > > >
> > > > We sincerely thank you for your efforts to improve the quality of our paper. Following your advice, we have further refined Figure 2. In the revised version,
> > > > * we add a sequential order to subfigure A to clarify the logic.
> > > > * we have divided subfigure C into three panels and provided detailed captions for each.
> > > >
> > > > Thank you again for your valuable contribution to our paper. We truly appreciate it.

---

> > > > > ### Comment · Reviewer_Qfc3 · 2024-11-25
> > > > > **Response to Further refined Figure 2**
> > > > >
> > > > > I thank the authors for their continued work in the improvement of the paper. The updates to the figure still leave me with residual questions: there is no clear identified starting or ending point, arrows of different shapes, sizes and colors are scattered throughout but are often pointing in the opposite direction of their counterparts without explanation.
> > > > >
> > > > > I will remain with my score of 6; the mixed results on practical testing, the lack of clarity on the diagrams (which are central to the explanation of the method are the primary concerns holding the score to a 6 as compared to an 8.

---

> > > > > > ### Author Response · Authors · 2024-11-26
> > > > > >
> > > > > > Thank you for your valuable comment, which has helped us improve further. We carefully considered your suggestion to adjust the layout of Fig. 2 in the revised version.
> > > > > >
> > > > > > In the updated Fig. 2, we applied a logical flow of “Panel 1” → “Panel 2” → “Panel 3” to clearly illustrate the workflow. Additionally, we removed several redundant elements to make the figure more concise and logical.
> > > > > >
> > > > > > We sincerely appreciate your contribution to enhancing the quality of our paper.

---

> > > > > > > ### Comment · Reviewer_Qfc3 · 2024-12-01
> > > > > > > **Acknowledgement of Update**
> > > > > > >
> > > > > > > I appreciate the work that the authors have put into the submission. The updates to the figure 2 create a noticeable improvement. I think over the course of the discussion period the strength of the submission has improved from just making a 5 to being a strong 6. However, I still feel that the mixed practical results and issues with the long tailed distribution, etc hold it back from being an 8. I view the core of the research as being very strong, but still a little rough and in need of refinements.
> > > > > > >
> > > > > > > I wish the authors the best of luck with the research, and I look forward to a published version that I can share with my collaborators one day.

---

> > > > > > > > ### Author Response · Authors · 2024-12-02
> > > > > > > > **Thank you**
> > > > > > > >
> > > > > > > > Dear Reviewer Qfc3:
> > > > > > > >
> > > > > > > > Thanks for your wishes. We sincerely appreciate your valuable contribution to the manuscript.
> > > > > > > >
> > > > > > > > Authors.

---

### Official Review · Reviewer_Z7kM · 2024-11-02

**Soundness:** 3
**Presentation:** 4
**Contribution:** 3
**Rating:** 6
**Confidence:** 3

**Summary:**

The paper introduces a benchmark dataset "SimXRD" which comprises 4,065,346 simulated powder XRD patterns that represent 119,569 unique crystal structures. This dataset aims to advance crystallographic informatics by providing high-quality training data for machine learning models. The authors evaluated 21 sequence models on their XRD dataset, testing both in-library and out-of-library classification scenarios. Their analysis revealed that current models struggle with rare crystal types and showed how different physical conditions affect model performance. The results demonstrate that models trained on simulated data can work with real XRD patterns.

**Strengths:**

This is essentially a benchmark paper that introduces a dataset that can be a useful resource for the community.

1. Creates the largest open-source XRD dataset (4M+ patterns) with validation against experimental data, enabling advancement in crystallographic machine learning research
2. Provides benchmarking across 21 models (CNNs, RNNs, Transformers) with detailed performance metrics
3. Tests both in-library and out-of-library scenarios and also evaluates against real experimental data (RRUFF dataset)
4. Identifies key challenges like long-tail distribution in crystal structures, providing directions for future machine learning model development

**Weaknesses:**

1. The paper benchmarks existing approaches without introducing new architectures to address the identified challenges, particularly for low-frequency crystal classification. This limits its technical novelty from an architectural perspective.

2. While generating experimental data is challenging, the validation against real-world data is limited. The RRUFF dataset (~3K patterns) represents only 0.07% of the simulated dataset size (4M+ patterns), and the comparison between simulated and real-world patterns is demonstrated only for Li2MnO3. A more comprehensive analysis of sim and real world gaps would be helpful.

3. The paper lacks investigation of model failure modes and interpretability analysis. For example, analyzing what patterns CNNs learn and how they make decisions could reveal important insights and potentially guide future architectural improvements. Such analysis would provide valuable direction for developing more effective models.

**Questions:**

1. Model Interpretability: Could you provide analysis of what patterns your best performing models learn from XRD data, and what features differentiate successful classifications from failures, especially for low-frequency crystals?

2. While not very important, it would be helpful to understand systematic differences between simulated and real XRD patterns beyond the Li2MnO3 case study, and identify which simulation parameters most affect model performance on real data.

---

> ### Author Response · Authors · 2024-11-20
> **Author Responses (1/2)**
>
> Thank you for your insightful feedback, which has significantly improved the quality of our manuscript. We have carefully addressed each of your comments, as detailed below. We greatly appreciate your recognition of the importance of this work in advancing phase identification. If you find our revisions satisfactory, we kindly request your favorable consideration for an improved score.
>
> >Weaknesses 1: Limit technical novelty from an architectural perspective.
>
> Thanks for your comment. We would like to clarify that our goal is to propose novel datasets and benchmarks (datasets and benchmarks track), which are also crucial besides architectural contributions. In this work, we provide a comprehensive and accessible powder XRD database, along with a benchmark designed to standardize symmetry identification tasks. We emphasize our technical contributions as follows:
>
> * We identify and highlight the heavy long-tailed distribution in symmetry classification, which has not been explored before and is vital for developing accurate models.
> * Through evaluating 21 model architectures, we find that: (1) Most existing CNN models are unsuitable for symmetry identification within the large scale of the SimXRD database; (2) Convolutional neural networks without pooling have achieved the best performance in most tasks; (3) The raw transformer encounters difficulties in learning XRD patterns while PatchTST achieves significant performance improvement, demonstrating subsequence-level patches are beneficial to identify peaks.
> * Through model interpretability analysis(**lines:810-821**), we discovered that: CNN11 places greater emphasis on characteristic peaks, aligning closely with the traditional search-match method. In contrast, BiGRU selectively focuses on relatively strong peaks, progressively reducing the emphasis on weaker peaks based on their intensity.
>
> >Weaknesses 2: The validation against real-world data is limited. The RRUFF dataset represents only 0.07% of the simulated dataset size.
>
> Thanks for your comment. We would like to clarify that labeled experimental data are indeed highly valuable, which is also the motivation for constructing a simulated XRD dataset. At present, RRUFF is the largest open-sourced experimental dataset as far as we know. Models trained on SimXRD can generalize to RRUFF, demonstrating the significance of our datasets.
>
> >Weaknesses 2: The comparison between simulated and real-world patterns is demonstrated only for Li2MnO3. A more comprehensive analysis of sim and real-world gaps would be helpful.
>
> Thanks for the constructive comment. In the revision, we emphasize the challenges posed by limitations in experimental patterns (**lines 073–079**): (1) insufficient data to comprehensively test generative capabilities, as you highlighted, and (2) the difficulty of effectively utilizing small-scale experimental datasets for model tuning and transfer learning. Additionally, we have conducted a more comprehensive analysis of the gaps between simulated and real-world data. This includes a supplement in Appendix B.4 XRD Simulation(**lines 1033–1070**), which examines the influence of simulation parameters, and a new section in Appendix B.8 Sim2Real Gap (**lines 1182–1232**), which compares the fidelity of pattern generation with GSAS-II.
>
> >Weaknesses 3: The paper lacks investigation of model failure modes and interpretability analysis. For example, analyzing what patterns CNNs learn and how they make decisions could reveal important insights and potentially guide future architectural improvements.
>
> Thank you for your valuable suggestions. Based on your advice, we have added a new section, Appendix A.3 Feature Analysis (**lines 792–821**), in the revised manuscript. The key observations are summarized as follows:
>
> * Machine learning models primarily rely on high-intensity peaks (characteristic peaks) to infer symmetry categories.
> * Models with better performance tend to extract more detailed information from relatively weak-intensity peaks, rather than relying solely on the "three strongest diffraction peaks" as in the search-match approach.
> * CNN11 places greater emphasis on characteristic peaks, aligning closely with the traditional search-match method. In contrast, BiGRU selectively focuses on relatively strong peaks, progressively reducing the emphasis on weaker peaks based on their intensity.

---

> > ### Author Response · Authors · 2024-11-20
> > **Author Responses (2/2)**
> >
> > >Questions 1: Could you provide analysis of what patterns your best performing models learn from XRD data, and what features differentiate successful classifications from failures, especially for low-frequency crystals?
> >
> > Thank you for your advice. In the revised Appendix A.3 Feature Analysis (**lines 792–821**), we analyze the features influencing the models' inferences and summarize the key observations as follows:
> >
> > * CNN11: This model places greater emphasis on characteristic peaks, closely aligning with the traditional search-match method.
> > * PatchTST:  PatchTST follows a similar trend, while capturing a broader range of peak characteristics compared to CNN11.
> > * BiGRU: BiGRU selectively focuses on relatively strong peaks, while progressively reducing emphasis on weaker peaks based on their intensity.
> >
> > In Appendix A.2 (**lines 746–788**), we categorize the XRD patterns by their frequency, revealing insights into the challenges of identifying low-frequency crystal systems/space groups. The conclusions are summarized below:
> > * Most models fail to predict XRD patterns of cubic and hexagonal symmetries. In contrast, several domain-specific CNN models (e.g., CNN 7, 9, 10, 11), along with LSTM, GRU, and advanced Transformers (e.g., iTransformer and PatchTST), can mitigate issues related to long-tail distributions.
> > * PatchTST demonstrates comparable performance to the best models across all low-frequency space groups.
> >
> > >Questions 2: It would be helpful to understand systematic differences between simulated and real XRD patterns beyond the Li2MnO3 case study, and identify which simulation parameters most affect model performance on real data.
> >
> > Thanks for your constructive comment.
> > * In Appendix B.4 (**lines 1033–1070**), we show how our simulation parameters influence a simulated PXRD pattern, making it more closely resemble an experimental one. The interactions between these parameters are illustrated in the formulas provided in Appendix B.4.
> > * In Table 6 (**lines 1135–1151**), we categorize the parameter space into three domains and offer insights into how these domains influence symmetry identification. Qualitative results indicate that the model struggles to accurately classify cases with significant peak broadening, which remains a challenge.
> > * In Appendix B.8 Sim2Real Gap (**lines 1182–1232**), we compare the fidelity of pattern generation with GSAS-II, demonstrating that our simulation tool is capable of generating high-fidelity simulations.

---

> > ### Comment · Reviewer_Z7kM · 2024-11-24
> > **Response to the author rebuttals**
> >
> > I thank the authors for the work and addressing my comments. I appreciate the work put in by the authors to improve the paper. My only concern that remains given the fact RRUFF dataset represents only 0.07% of the simulated dataset size. Can we claim generalizability based on the good performance on that dataset. In any case, I don't think it is possible to address my concerns unless a larger experimental dataset becomes available.

---

> > > ### Author Response · Authors · 2024-11-24
> > >
> > > We sincerely thank you for the comment. In the revised version, we have made four changes to clarify the claim to make it more accurate:
> > >
> > > * (Lines 75-76) We clarified that, due to the limitations of experimental patterns, it remains challenging to comprehensively test the generative capabilities of models.
> > > * (Lines 107) We stated that the conclusions regarding experimental data are based on tests conducted using the RRUFF dataset.
> > > * (Lines 473) In the benchmark results for generalization testing, we noted that the results were derived from the RRUFF dataset.
> > > * (Lines 1290-1338) A new section (Appendix B.10) was added to compare the XRD distributions from SimXRD and RRUFF, providing insights into their differences and consistencies.
> > >
> > > We sincerely appreciate your support for our paper and your encouragement in fostering progress in this field.

---

> > > > ### Comment · Reviewer_Z7kM · 2024-12-03
> > > >
> > > > I would like to thank the authors for their clarifications. In my assessment, a solid 6 appropriately reflects my evaluation of this work.

---

> > > > > ### Author Response · Authors · 2024-12-04
> > > > >
> > > > > Dear Reviewer,
> > > > >
> > > > > Thank you for your valuable contributions to this paper. Your feedback has greatly enhanced the quality of the manuscript, and we sincerely appreciate your efforts.

---

### Meta-Review · Area_Chair_5jD9 · 2024-12-13

**Metareview:**

This submission to the datasets and benchmarks track introduces a new simulated high-fidelity X-ray diffraction (XRD) pattern dataset, SimXRD, which can help benchmark and further develop machine learning (ML) methods in crystallographic informatics. The simulation method for SimXRD incorporates comprehensive physical interactions, with 4,065,346 simulated powder XRD patterns from 119,569 unique crystal structures under 33 simulated conditions, which is claimed to be the largest open-source XRD dataset. The authors also benchmarked existing ML methods related to data-imbalance effect and showed that training models with SimXRD can be generalized to experimental data.

The reviewers agree that SimXRD can contribute to ML development of crystallographic informatics. The benchmarking results provide insights and potential research directions in analyzing XRD patterns for crystal identification and materials discovery.

**Additional Comments On Reviewer Discussion:**

During the rebuttal discussion, the authors have provided additional analysis results and explanations to improve the presentation and clarify their contributions as well as the indication from ML benchmark results.

Some reviewers after rebuttal discussion still have concerns on the presentation, novelty, and the generalizability of this simulated benchmark to real-world experimental data.

---

### Decision · Program_Chairs · 2025-01-22

Accept (Poster)